# A biophysical model of striatal microcircuits suggests gamma and beta oscillations interleaved at delta/theta frequencies mediate periodicity in motor control

Julia A. K. Chartove[1]*, Michelle M. McCarthy[2], Benjamin R. Pittman-Polletta[2], Nancy J. Kopell[2]

**1** Graduate program in Neuroscience, Center for Systems Neuroscience, Boston University, Boston, Massachusetts, United States of America, **2** Department of Mathematics & Statistics, Boston University, Boston, Massachusetts, United States of America

☯ These authors contributed equally to this work.
* chartove@bu.edu

**Data Availability Statement:** All code used to generate the simulations and figures in this manuscript is available at

## Abstract

Striatal oscillatory activity is associated with movement, reward, and decision-making, and observed in several interacting frequency bands. Local field potential recordings in rodent striatum show dopamine- and reward-dependent transitions between two states: a "spontaneous" state involving $\beta$ ($\sim$15-30 Hz) and low $\gamma$ ($\sim$40-60 Hz), and a state involving $\theta$ ($\sim$4-8 Hz) and high $\gamma$ ($\sim$60-100 Hz) in response to dopaminergic agonism and reward. The mechanisms underlying these rhythmic dynamics, their interactions, and their functional consequences are not well understood. In this paper, we propose a biophysical model of striatal microcircuits that comprehensively describes the generation and interaction of these rhythms, as well as their modulation by dopamine. Building on previous modeling and experimental work suggesting that striatal projection neurons (SPNs) are capable of generating $\beta$ oscillations, we show that networks of striatal fast-spiking interneurons (FSIs) are capable of generating $\delta/\theta$ (ie, 2 to 6 Hz) and $\gamma$ rhythms. Under simulated low dopaminergic tone our model FSI network produces low $\gamma$ band oscillations, while under high dopaminergic tone the FSI network produces high $\gamma$ band activity nested within a $\delta/\theta$ oscillation. SPN networks produce $\beta$ rhythms in both conditions, but under high dopaminergic tone, this $\beta$ oscillation is interrupted by $\delta/\theta$-periodic bursts of $\gamma$-frequency FSI inhibition. Thus, in the high dopamine state, packets of FSI $\gamma$ and SPN $\beta$ alternate at a $\delta/\theta$ timescale. In addition to a mechanistic explanation for previously observed rhythmic interactions and transitions, our model suggests a hypothesis as to how the relationship between dopamine and rhythmicity impacts motor function. We hypothesize that high dopamine-induced periodic FSI $\gamma$-rhythmic inhibition enables switching between $\beta$-rhythmic SPN cell assemblies representing the currently active motor program, and thus that dopamine facilitates movement in part by allowing for rapid, periodic shifts in motor program execution.

https://github.com/jchartove/striatum-standalone. A persistent copy of the model is also available at ModelDB as "Striatal FSI and SPN oscillation model (Chartove et al. 2020)" (https://senselab. med.yale.edu/ModelDB/showmodel.cshtml? model=261461). All other relevant data are within the paper and its Supporting Information files.

**Funding:** This work was funded by National Science Foundation Division of Mathematical Sciences (https://www.nsf.gov/) grant DMS-1514796 (PI: NJK) and National Institute of Mental Health (https://www.nimh.nih.gov/index.shtml) grant 1R01MH114877-01 (PI: NJK). The funders had no role in study design, data collection and analysis, decision to publish, or preparation of the manuscript.

**Competing interests:** The authors have declared that no competing interests exist.

## Author summary

Striatal oscillatory activity is associated with movement, reward, and decision-making, and observed in several interacting frequency bands. The mechanisms underlying these rhythmic dynamics, their interactions, and their functional consequences are not well understood. In this paper, we propose a biophysical model of striatal microcircuits that comprehensively describes the generation and interaction of striatal rhythms, as well as their modulation by dopamine. Our model suggests a hypothesis as to how the relationship between dopamine and rhythmicity impacts the function of the motor system, enabling rapid, periodic shifts in motor program execution.

## Introduction

As the largest structure of the basal ganglia network, the striatum is essential to motor function and decision making. It is the primary target of dopaminergic (DAergic) neurons in the brain, and its activity is strongly modulated by DAergic tone. Disorders of the DA and motor systems, such as Parkinson's, Huntington's, Tourette's, and many others, result in abnormal network activity within striatum [1–9]. Rhythmic activity is observed in both striatal spiking and local field potential, and oscillations in the striatum are correlated with voluntary movement, reward, and decision-making in healthy individuals [10–18], while disruptions of these rhythms are biomarkers of mental and neurological disorders [1, 2, 19–27]. However, the mechanisms of these oscillations, and their role in motor behavior and its dysfunctions, remain poorly understood.

The current study focuses on the oscillatory bands frequently observed in striatal local field potential: $\delta$ (1-3 Hz), $\theta$ (4-7 Hz), $\beta$ (8-30 Hz), low $\gamma$ (50-60 Hz), and high $\gamma$ (70-80 Hz) [10, 16, 28]. Power in these bands consistently correlates with responses to task parameters including motor initiation, decision making, and reward [10–12, 20]. Power in the $\beta$ band is elevated in Parkinson's disease and correlates with the severity of bradykinesia [2], while striatal $\gamma$ is associated with the initiation and vigor of movement [18, 20]. In the healthy basal ganglia, $\beta$ and $\gamma$ activity are inversely correlated and differentially modulated by slower basal ganglia rhythmic activity, suggesting that the balance of these distinct oscillatory dynamics is important to healthy motor function [16]. In rat striatum *in vivo*, spontaneous $\beta$ and low $\gamma$ oscillations transition to $\theta$ and high $\gamma$ dynamics upon reward receipt and with administration of DA agonist drugs [10]; similarly, in rat caudate and putamen, DAergic agonists produce robust low-frequency modulation of high $\gamma$ amplitude [28].

In this paper, we propose a biophysical model of striatal microcircuits that comprehensively describes the generation and interaction of these rhythms, as well as their modulation by DA. Our simulations capture the dynamics of networks of striatal fast-spiking interneurons (FSIs) and striatal projection neurons (SPNs), using biophysical Hodgkin-Huxley type models. Our model consists of three interconnected populations of single or double compartment Hodgkin-Huxley neurons: a feedforward network of FSIs, and two networks of SPNs (the D1 receptor-expressing "direct pathway" subnetwork and the D2 receptor-expressing "indirect pathway" subnetwork). SPNs, responsible for the output of the striatum, make up 95% of striatal neurons in rodents [29]. SPN firing is regulated by relatively small populations of striatal interneurons, including fast spiking interneurons (FSIs), which strongly inhibit SPNs. Our model FSIs exhibit a D-type potassium current [30], and our model SPNs exhibit an M-type potassium current [31]. Both cell types are modulated by DAergic tone: FSIs express the excitatory D1 DA receptor [32], while two distinct subpopulations of SPNs express exclusively the

D1 or the inhibitory D2 receptor subtype. We modeled both SPN subpopulations, with high simulated DAergic tone increasing and decreasing D1 and D2 SPN excitability, respectively. To model DA effects on the FSI network, we simulated three experimentally observed effects: increased excitability due to depolarization [32], increased gap junction conductance [33], and decreased conductance of inhibitory synapses [32]. Both gap junctions and inhibition are known to play a role in the generation of rhythmic activity [34–44].

Our previous experimental and modeling work suggests that striatal SPN networks can produce a $\beta$ (15-25 Hz) oscillation locally [45]. Our current model demonstrates that FSI networks can produce $\delta/\theta$ (~3-6 Hz), low $\gamma$, and high $\gamma$ oscillations. A fast-activating, slow-inactivating potassium current (the D-type current) allows FSIs to produce $\gamma$ and $\delta/\theta$ rhythms in isolation, and network interactions make these rhythms, otherwise highly susceptible to noise, robust. In our simulations, DA induces a switch between two FSI network states: a low DA state exhibiting persistent low $\gamma$ rhythmicity, and a high DA state in which a $\delta/\theta$ oscillation modulates high $\gamma$ activity. As a result of FSI inhibition of SPNs, DA induces a switch in striatal dynamics, between a low DA state in which low $\gamma$ and $\beta$ rhythms coexist, and a high DA state in which bursts of FSI-mediated high $\gamma$ and SPN-mediated $\beta$ rhythms alternate, nested within (and appearing at opposite phases of) an FSI-mediated $\delta/\theta$ rhythm. Thus, our model generates a hypothesis as to how observed relationships between DA and rhythmicity impact the function of the motor system. Namely, DA appears to encourage or permit periodic motor program switching, allowing the emergence of an FSI-mediated $\delta/\theta$-nested $\gamma$ rhythm, which in turn breaks up the "stay" signal mediated by SPN $\beta$ rhythms [46].

## Results

### Single model FSIs produce $\delta/\theta$-nested $\gamma$ rhythms whose power and frequency is modulated by excitation

We modified a previous single-compartment striatal FSI model [47] by adding a dendritic compartment (shown to be an important determinant of gap-junction mediated synchrony [48–51]) and increasing the conductance of the D-type K current to 6 $mS/cm^2$. Previous work showed that two characteristic attributes of FSI activity *in vitro*, stuttering and $\gamma$ resonance (defined as a minimal tonic firing rate in the $\gamma$ frequency range), are dependent on the D-current [30, 47]. Our modified FSI model successfully reproduced these dynamics as well as revealing other dynamical behaviors (Fig 1).

With increasing levels of tonic applied current ($I_{app}$), our model FSI transitions from quiescence to (periodic) bursting to periodic spiking. The bursting regime, of particular interest in this work, is dependent on the level of tonic excitation and, centrally, the D-current conductance (Fig 1). FSI spiking frequency increases with tonic drive (Fig 1A). As shown previously [47], the FSI model's $\gamma$-rhythmic intraburst spiking arises from its minimum firing rate, which is also set by the D-current conductance. When this conductance is zero, the model has no minimum firing rate; firing rate is a continuous function of $I_{app}$ with a minimum firing rate of zero (Fig 1B). As the D-current conductance is increased, the firing rate below which the cell will not fire also increases. Therefore, our choice of D-current ($g_d$ = 6, resulting in a minimum firing rate around 40 Hz) reflects not only our interest in the bursting regime, but also our desire to match experimental observations of striatal $\gamma$ frequency [10, 47].

The frequency of bursting depends on the decay time constant of the D-type potassium current ($\tau_D$); in the absence of noise, it is in the $\delta$ frequency range for physiologically relevant $\tau_D$ ($< \sim 200$ ms, Fig 1C). Note that $\tau_D$ changes the inter-burst interval without changing the timing of spikes within a burst. With lower levels of D-current (as used in previous FSI models [30, 47, 52]), bursting is aperiodic. For sufficiently large D-current conductance, FSI bursting

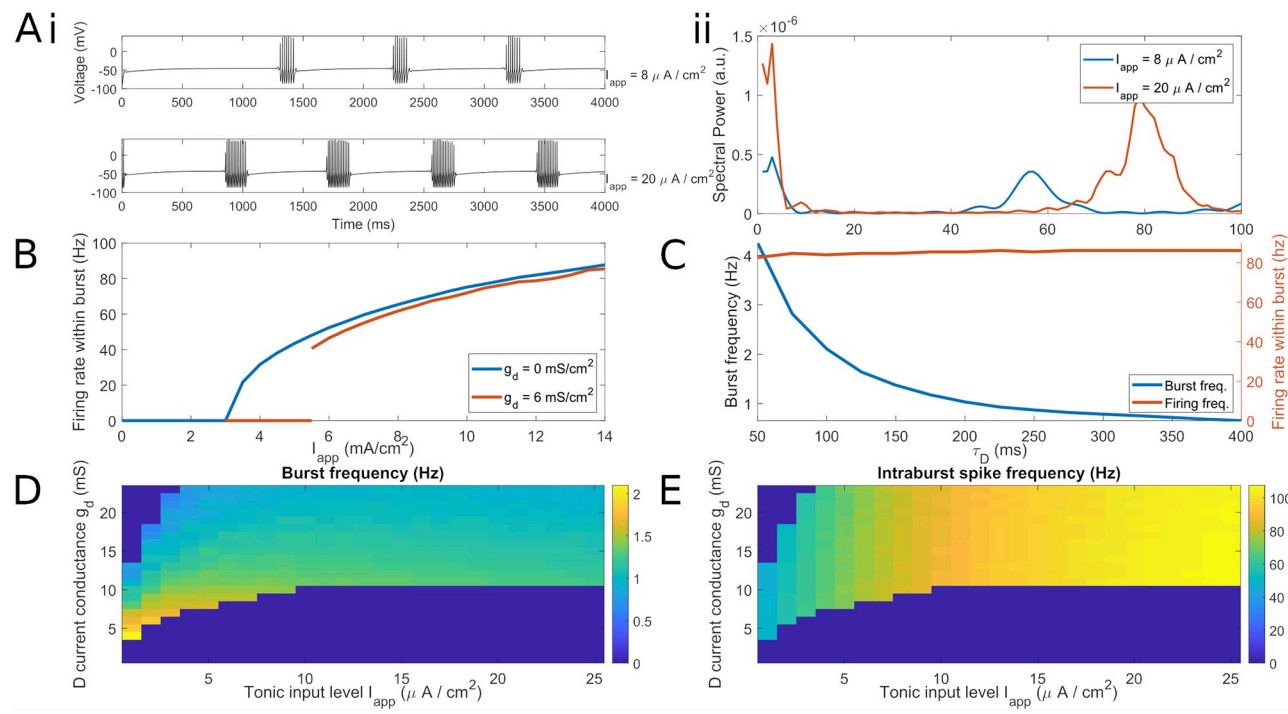

**Fig 1. Behavior of single model FSI over a range of applied currents and D-current conductances.** (A) i. A single model FSI with low tonic excitation ($I_{app} = 8\mu A/cm^2$) spikes at a low $\gamma$ frequency within periodic bursts, while a single model FSI with high tonic excitation ($I_{app} = 20\mu A/cm^2$) spikes at a high $\gamma$ within periodic bursts. ii. Power spectral density of voltage traces in (A)i, comparing low and high levels of tonic excitation. Power spectra are derived using Thomson's multitaper power spectral density (PSD) estimate (MATLAB function pmtm). (B) Plot of the minimal firing rate within a burst of a single model FSI with zero and nonzero D current conductance $g_D$. Note that the cell does not fire below 40 Hz when the D-current is present. (C) Plot of the maximal inter-burst ($\delta$) frequency and intraburst ($\gamma$) firing rate of a single model FSI as $\tau_D$, the time constant of inactivation of the D current, is increased. (D) Three-dimensional false-color plot demonstrating the dependence of the bursting regime on $g_d$ and $I_{app}$. (E) Three-dimensional false-color plot demonstrating the dependence of firing rate on $g_d$ and $I_{app}$.

occurs for a broad range of applied currents ($I_{app}$ over 5 $\mu A/cm^2$, Fig 1D and 1E). Since simulated DA acts on our FSI model by increasing tonic excitation, DA causes an increase in model FSI spiking from low $\gamma$ rhythmicity to high $\gamma$ rhythmicity. Below, we demonstrate that the FSI $\gamma$ is determined by this single-cell rhythmicity and is mostly independent of the timescale of inhibitory synapses.

In addition to increasing with tonic excitation, burst frequency increases to $\delta/\theta$ frequencies when the input includes small amounts of noise (Fig 2A and 2B), which decrease the interburst interval. However, noise of sufficient amplitude abolishes rhythmic bursting altogether (at least in single cells, Fig 2C).

In summary, a single model FSI displays low-frequency-nested $\gamma$ oscillations, dependent on the D-type current, under a wide range of tonic excitation levels. Both low frequency power and $\gamma$ frequency increase with tonic excitation. While noise increases the frequency of the slower rhythm from $\delta$ to $\theta$, it also diminishes the power of this rhythm in the single cell. Below we demonstrate that all of these effects are also present in a network of FSIs, with a key difference: the network $\delta/\theta$ rhythm is robust to noise.

## FSI networks produce DA-dependent $\delta/\theta$ and $\gamma$ rhythms

To determine if $\gamma$ and $\delta/\theta$ oscillations persist in networks of connected FSIs, and how DA could modulate these network dynamics, we simulated a network of 50 model FSIs connected randomly by both inhibitory synapses (connection probability 0.58 [53]) and gap junctions

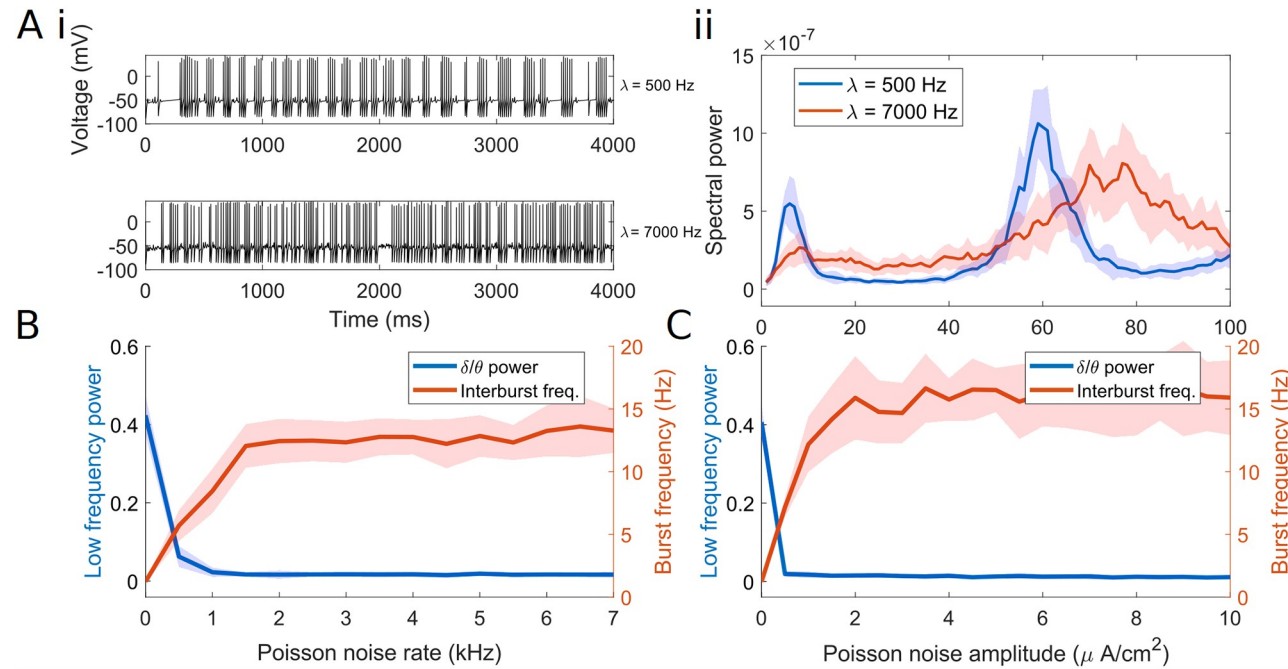

**Fig 2. Applied noise determines interburst and intraburst frequency of FSI spiking.** (A) i. Single model FSI with tonic excitation (7 $\mu A/cm^2$) and weak Poisson noise ($\lambda = 500$) spikes at $\gamma$ nested in $\delta/\theta$, while a single model FSI with tonic excitation (7 $\mu A/cm^2$) and strong Poisson noise ($\lambda = 7000$) has limited low-frequency content. ii. Power spectral density of voltage traces in (A)i, comparing low and high levels of noise. The solid line represents the mean value over 20 simulations per point. Shading represents standard deviation from these means. Power spectra are derived using Thomson's multitaper power spectral density (PSD) estimate (MATLAB function pmtm). (B) Plot of the inter-burst frequency and power of a single model FSI as Poisson noise of varying rates is applied. (C) Plot of the inter-burst frequency and power of a single model FSI as Poisson noise of varying amplitudes is applied. For B and C $I_{app} = 7\ \mu A/cm^2$.

(connection probability 0.33 [54]). We also implemented three experimentally observed effects of DA on FSI networks: increased tonic excitation of individual FSIs [32], increased gap junction conductance between FSIs [33], and decreased inhibitory conductance between FSIs [32] (see Methods). We used the sum of all synaptic inputs within the network as a surrogate measure for simulated local field potential (LFP); this measure is hereafter referred to as "surrogate LFP".

Unlike in single cells, FSI network $\delta/\theta$ rhythmicity is dependent on sufficient levels of tonic excitation: at low levels of tonic input ($I_{app} < \sim 1\mu A/cm^2$), the FSIs do not attain enough synchrony for a strong network $\delta/\theta$ (Fig 3Ai). As in single cells, FSI network $\delta/\theta$ power increases with tonic input strength (Fig 3Ai). Sufficiently strong gap junction coupling is also a requirement for the FSI network to attain sufficient synchrony to produce $\delta/\theta$ rhythmicity (Fig 3Bi). Gap junctions function to protect the FSI network $\delta/\theta$ rhythm from the effects of noise (as in [39, 55]); the $\delta/\theta$ oscillation in the network is far more robust to noise than the same oscillation in a single cell (S1 Fig). Finally, inhibitory synaptic interactions between FSIs have a desynchronizing effect that interferes with network $\delta/\theta$, and increasing inhibitory conductance within the FSI network decreases power in the $\delta/\theta$ band (Fig 3Ci). FSI network $\gamma$ power and frequency both increase with tonic input strength (Fig 3Aii), and, like the network $\delta/\theta$, the network $\gamma$ rhythm is dependent on sufficient gap junction conductance and is disrupted by inhibition (Fig 3Bii & 3Cii). Both network rhythms are robust to a wide range of heterogeneity in applied current and conductances (S2 Fig).

To explore whether the $\gamma$ rhythms observed in the FSI network are generated by inhibitory interactions, we examined the dependence of $\gamma$ frequency on the time constant of GABA$_A$

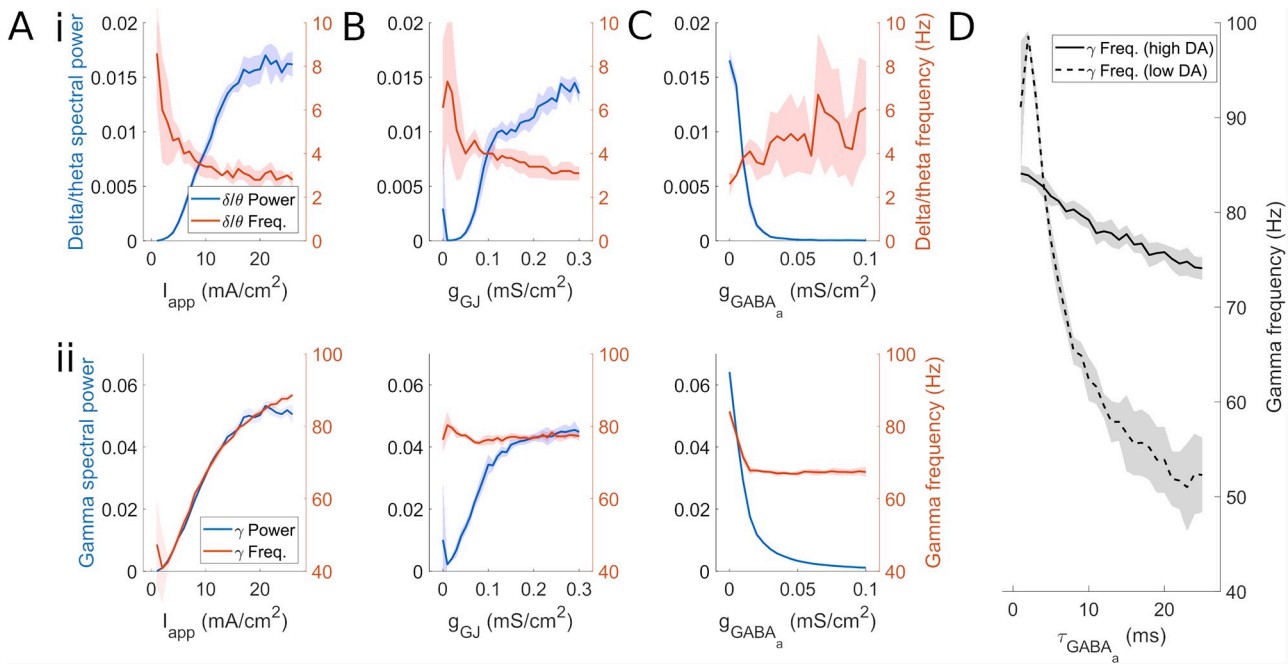

**Fig 3. FSI network rhythms change with background excitation and synaptic strength.** Power and frequency of $\delta/\theta$ and $\gamma$ rhythms in FSI network mean voltage as a function of (A) tonic input current, (B) gap junction conductance, and (C) GABA$_A$ conductance. The parameters not being varied in plots A-C are held at the high DA values ($I_{app}$ = 14 $\mu A/cm^2$, $g_{GJ}$ = 0.3 $mS/cm^2$, $g_{syn}$ = 0.005 $mS/cm^2$, $\tau_{gaba}$ = 13 ms. The solid line represents the mean value over 10 simulations per point. Shading represents standard deviation from these means. Power spectra are derived using Thomson's multitaper power spectral density (PSD) estimate (MATLAB function pmtm). (D) Gamma frequency as a function of GABA$_a$ synaptic time constant and level of dopamine. High DA values are as previously stated; low DA values are $I_{app}$ = 7 $\mu A/cm^2$, $g_{GJ}$ = 0.15 $mS/cm^2$, $g_{syn}$ = 0.1 $mS/cm^2$.

inhibition, as the characteristic frequency of canonical interneuron network $\gamma$ (ING) has been shown to depend on this time constant [38, 40, 56, 57]. The frequency of the $\gamma$ rhythm produced under low DA conditions decreased with increases in the GABA$_A$ time constant (Fig 3D), suggesting this rhythm is ING-like. However, the $\gamma$ produced under high DA conditions had a frequency that was not highly dependent on the inhibitory time constant, suggesting that this $\gamma$ rhythm is mechanistically different from previous ING models, being generated by synchronous $\gamma$ frequency bursts in individual cells, as opposed to inhibitory interactions.

In order to explore FSI network dynamics that might be observed during normal fluctuations in DA during goal-directed tasks [58], we simulated FSI network activity under two conditions, simulated low (or baseline) and high DAergic tone (Fig 4A). Parameter values for low and high DA were chosen so as to best demonstrate qualitative differences in network behaviors while maintaining physiologically realistic behavior on the cellular level (see Methods).

During simulated low DAergic tone, characterized by low levels of FSI tonic excitation and gap junction conductance, and high levels of inhibitory conductance, the network produces a persistent low frequency $\gamma$ oscillation ($\sim$ 60 Hz) in the surrogate LFP (Fig 4Bi–4Di). The raster plot of FSI spike times (Fig 4Eii) shows that individual FSIs exhibit sparse spiking in the low DA state. Although individual FSIs exhibit periodic spike doublets or bursts ($\gamma$-paced and entrained to the network $\gamma$) that recur at $\delta/\theta$ frequency, the timing of these bursts is independent (Fig 4Di and 4Ei). Therefore, while $\delta/\theta$ power is present at the level of individual FSIs, there is not sufficient synchrony for it to appear in the network; while the voltages of individual cells show power in the $\delta/\theta$ band, a power spectrum of the surrogate LFP does not (Fig 4Di).

During simulated high DAergic tone, characterized by high levels of tonic excitation and gap junction conductance and low levels of inhibitory conductance, network activity is

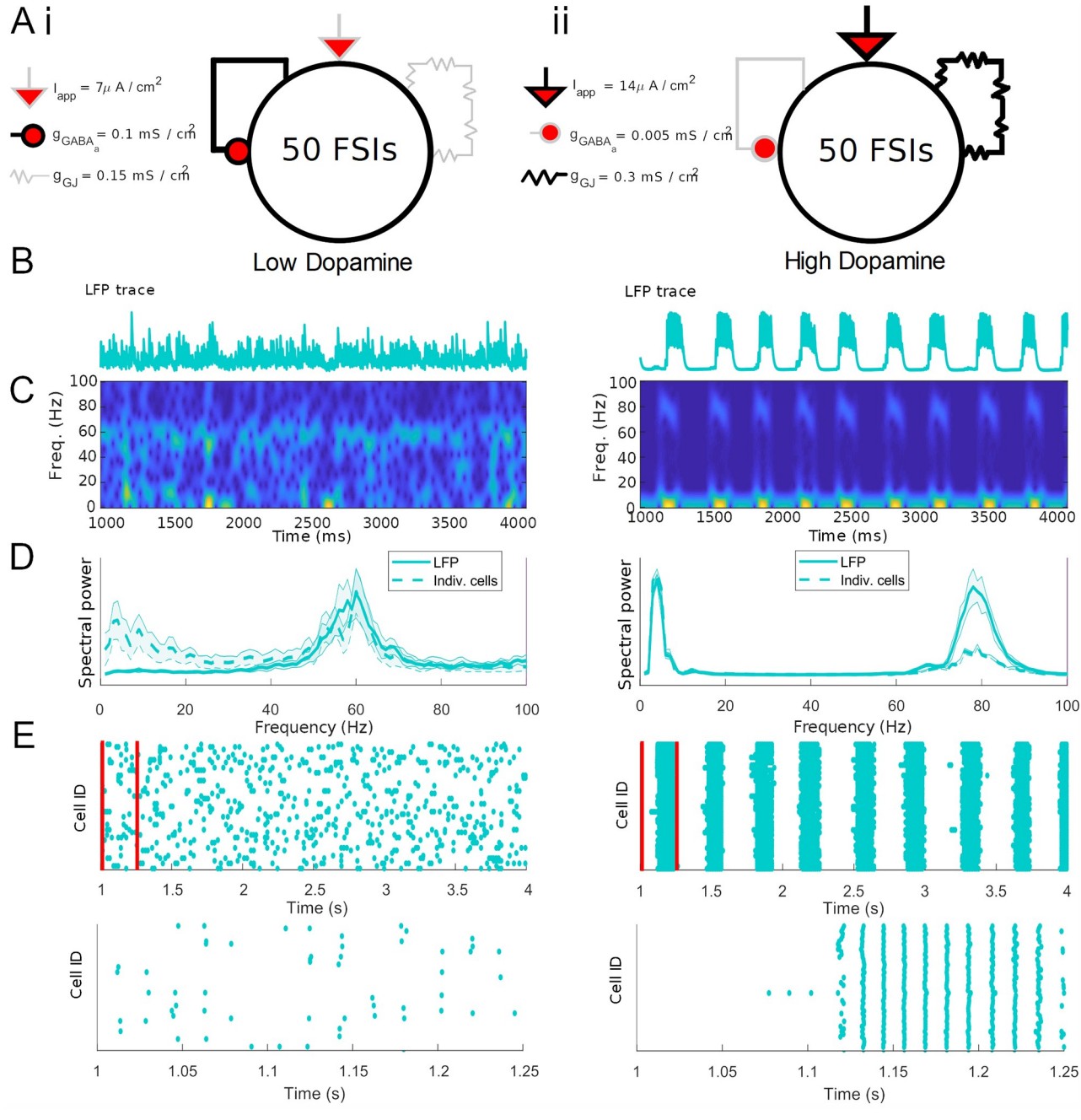

**Fig 4. FSI network activity and rhythms are altered by DA.** (A) Schematics showing the effects of dopamine on the FSI network during the baseline (i) and high (ii) DAergic tone conditions. (B) Sum of synaptic currents (surrogate LFP) for the FSI network in the two conditions. (C) Spectrograms of (B). (D) Solid line: Power spectral density of summed FSI synaptic currents (surrogate LFP), averaged over 20 simulations. Dashed line: Average power spectral density of each individual FSI voltage trace in the network, averaged over 20 simulations. Shading represents standard deviation from the mean. (E) Raster plots of FSI network activity at multisecond and subsecond timescales (red bars indicate time limits of lower raster plot).

much more structured: a strong 80 Hz $\gamma$ rhythm, phase-modulated by a 3 Hz $\delta/\theta$ rhythm, is evident in both the surrogate LFP and network raster plots (Fig 4Bii–4Eii, right). In this state, active FSIs spike at the same phase of both $\delta/\theta$ and $\gamma$, producing dual (and nested) network rhythms.

## SPN networks generate DA-dependent $\beta$ oscillations

Previous work by our group on the striatal origin of pathological oscillations in Parkinson's disease found that robust $\beta$ oscillations can emerge from inhibitory interactions in model networks of SPNs [45]. The interaction of synaptic GABA$_A$ currents and intrinsic M-currents promotes population oscillations in the $\beta$ frequency range; their $\beta$ timescale is promoted by the M-current, which allows rebound excitation at $\sim 50$ ms in response to synaptic inhibition. Excitation of SPNs increases $\beta$ power and frequency (see Methods). With this previous striatal SPN network model, we explored the transition from a healthy to a parkinsonian state with pathologically low levels of striatal DA [45]. Here, to explore the generation of $\beta$ rhythmicity during normal fluctuations in DAergic tone, we simulated two independent networks of 100 D1 receptor expressing ("direct pathway") SPNs and 100 D2 receptor expressing ("indirect pathway") SPNs. Model SPNs are single compartment neurons expressing the Hodgkin-Huxley spiking currents and the M-type potassium current, interconnected all-to-all by weak inhibitory GABA$_A$ synapses (i.e., connection probability 1). We simulated the effects of DA on model D1 and D2 SPNs by increasing and decreasing their levels of tonic excitation, respectively. (Whether DA generates a positive or negative applied current was the only difference between D1 and D2 expressing SPNs in our model; see Methods and Fig 5A. For further explanations of parameter choices and discussion of simplifications made while modeling the network, see the "Caveats and limitations" section of the Discussion.) In the absence of FSI input, neither population was sufficiently excited to exhibit spontaneous spiking under low DA conditions (Fig 5i). Subthreshold low-$\beta$ oscillations are present in the mean voltage of the non-firing SPN networks due to the timescale of the M-type potassium current [45]. Under high DA conditions, D1 SPNs exhibited persistent high-$\beta$ rhythmicity at $\sim 20$ Hz (Fig 5ii) due to the increase in applied current.

## FSI network $\gamma$ and $\delta/\theta$ oscillations rhythmically modulate SPN network $\beta$ oscillations only in high DA state

To understand the interactions between FSI and SPN networks, and between $\beta$, $\gamma$, and $\delta/\theta$ rhythms, we simulated a combined FSI-SPN striatal microcircuit, in which 50 model FSIs randomly connect to two independent networks of 100 SPNs, one each consisting of D1 and D2 SPNs (connection probability from FSIs to D1 or D2 SPNs of 0.375 [52]). FSIs were interconnected by gap junctions and inhibitory synapses (connection probability 0.33 and 0.58 respectively). D1 and D2 SPNs were connected by all-to-all inhibitory synapses (connection probability 1) within but not across populations. There were no connections from SPNs back to FSIs [59].

During simulated baseline DAergic tone, we modeled D1 and D2 SPNs as being equally excitable, with equal firing rates matching *in vivo* observations [60] while under the influence of FSI inhibition. The presence of FSIs is sufficient for the SPNs to fire in the low dopamine state (Fig 6i); this paradoxical excitatory effect of GABAergic input arises because SPNs can be excited via post-inhibitory rebound, as demonstrated in previous work [45]. Both SPN networks produce a low-$\beta$ rhythm (15 Hz), while the FSI network produces a low $\gamma$ (60 Hz, Figs 6i & 7i). The SPN subnetwork does not entrain to the FSI $\gamma$. The generation of low $\gamma$ and $\beta$ rhythms matches observations of striatal rhythmicity in resting healthy animals *in vivo* [10]. Our model suggests that these $\gamma$ and $\beta$ rhythms are independently generated by FSI and SPN networks, respectively.

During simulated high DAergic tone, an FSI-mediated high $\gamma$ ($\sim 80$ Hz) and an SPN-mediated $\beta$ ($\sim 15$-20 Hz) are observed during opposite phases of an ongoing FSI network $\delta/\theta$ rhythm (Figs 6ii & 7ii). During the peak of the $\delta/\theta$, the incoming $\gamma$ frequency input from the

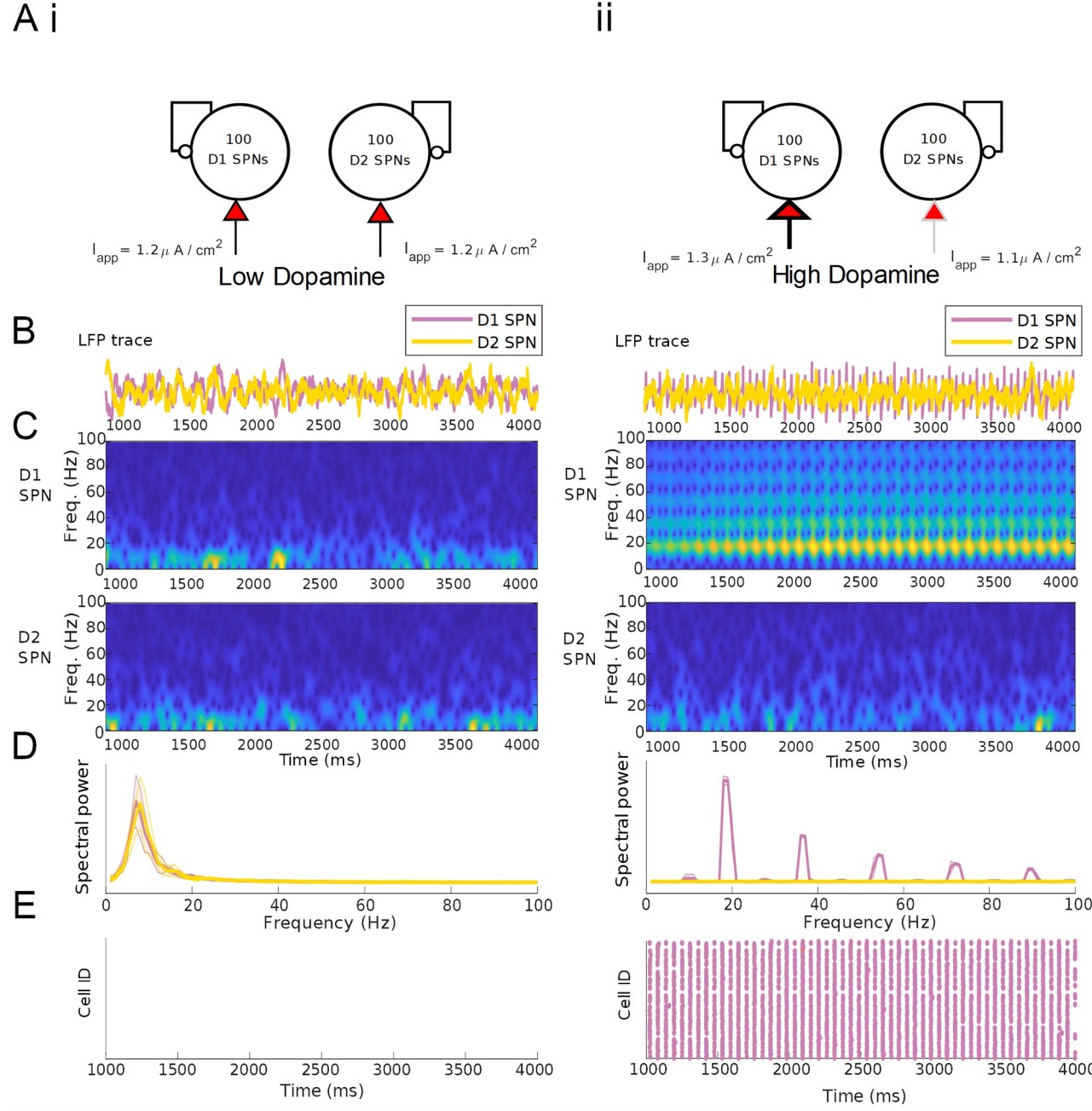

**Fig 5. Baseline SPN activity is characterized by β oscillations only in the D1 subnetwork under high DA conditions.** (A) Schematics depicting the baseline (i) and high DAergic tone (ii) conditions in an isolated SPN-only network. (B) Mean voltages for the D1 and D2 SPN populations in the two conditions. (C) Spectrograms of mean voltage for the D1 subpopulation (upper) and D2 subpopulation (lower). (D) Power spectral density of D1 and D2 population activity, averaged over 20 simulations. Shading represents standard deviation from the mean. Power spectra are derived using Thomson's multitaper power spectral density (PSD) estimate (MATLAB function pmtm). (E) Raster plots of SPN population activity.

FSIs silences the SPNs. When the FSIs are silent during the $\delta/\theta$ trough, both D1 and D2 SPN populations are sufficiently excited to produce a $\beta$ rhythm. Thus, while the SPNs cannot entrain to the $\gamma$ frequency of FSI inhibition, they are modulated by the FSI-generated $\delta/\theta$ rhythm. Due to the differences in excitability under high DAergic tone, the D1 SPN subpopulation produces a higher frequency $\beta$ ($\sim 20$ Hz) than does the less excitable D2 subpopulation,

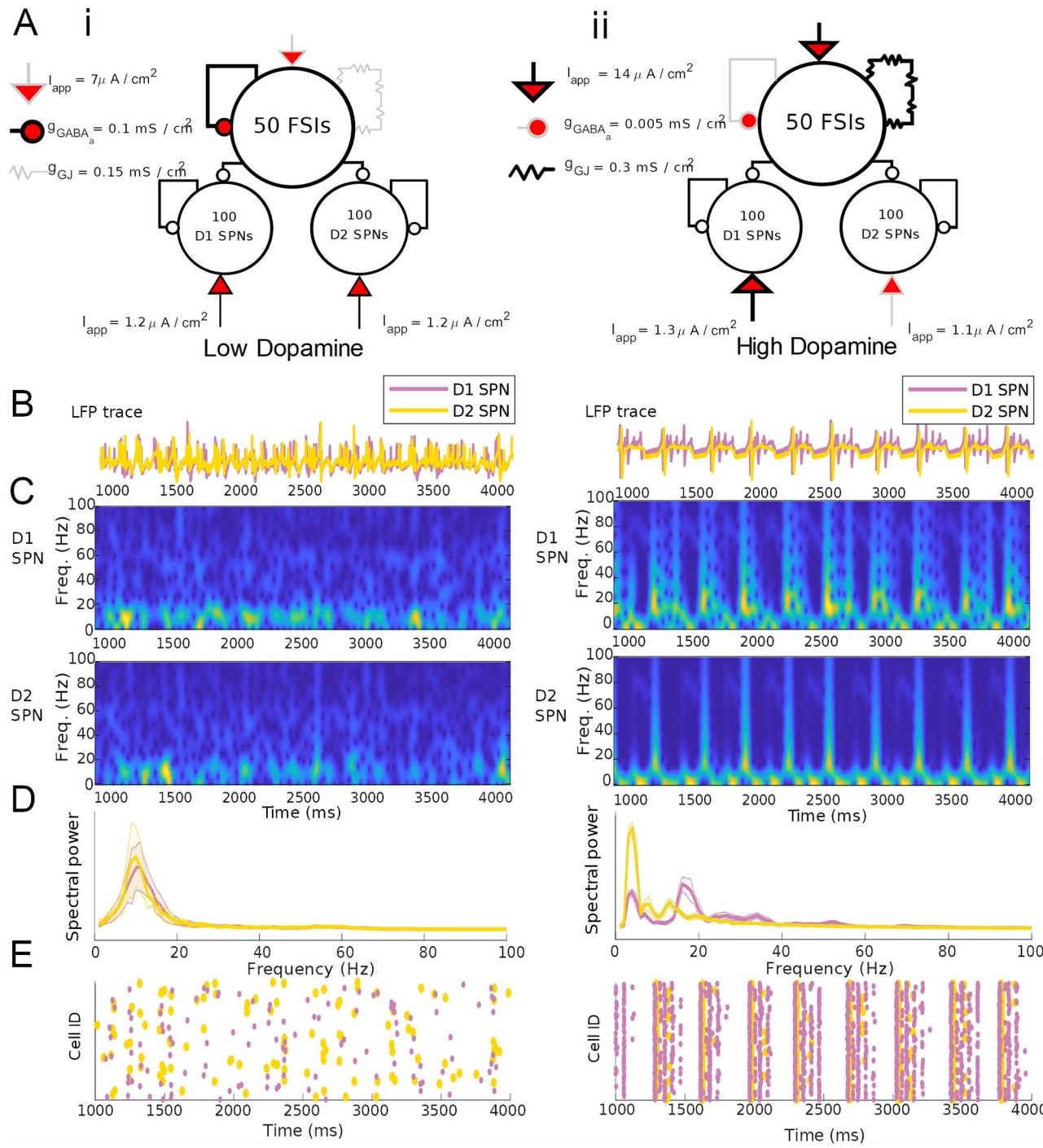

**Fig 6. FSIs paradoxically excite and pattern SPN network activity.** (A) Schematics showing modulation during the baseline (i) and high (ii) DAergic tone conditions in a combined FSI-SPN network. (B) Mean voltages for the D1 and D2 SPN populations in the two conditions. (C) Spectrograms of mean voltage for the D1 subpopulation (upper) and D2 subpopulation (lower). (D) Power spectral density of D1 and D2 population activity, averaged over 20 simulations. Shading represents standard deviation from the mean. Power spectra are derived using Thomson's multitaper power spectral density (PSD) estimate (MATLAB function pmtm). (E) Raster plots of SPN population activity.

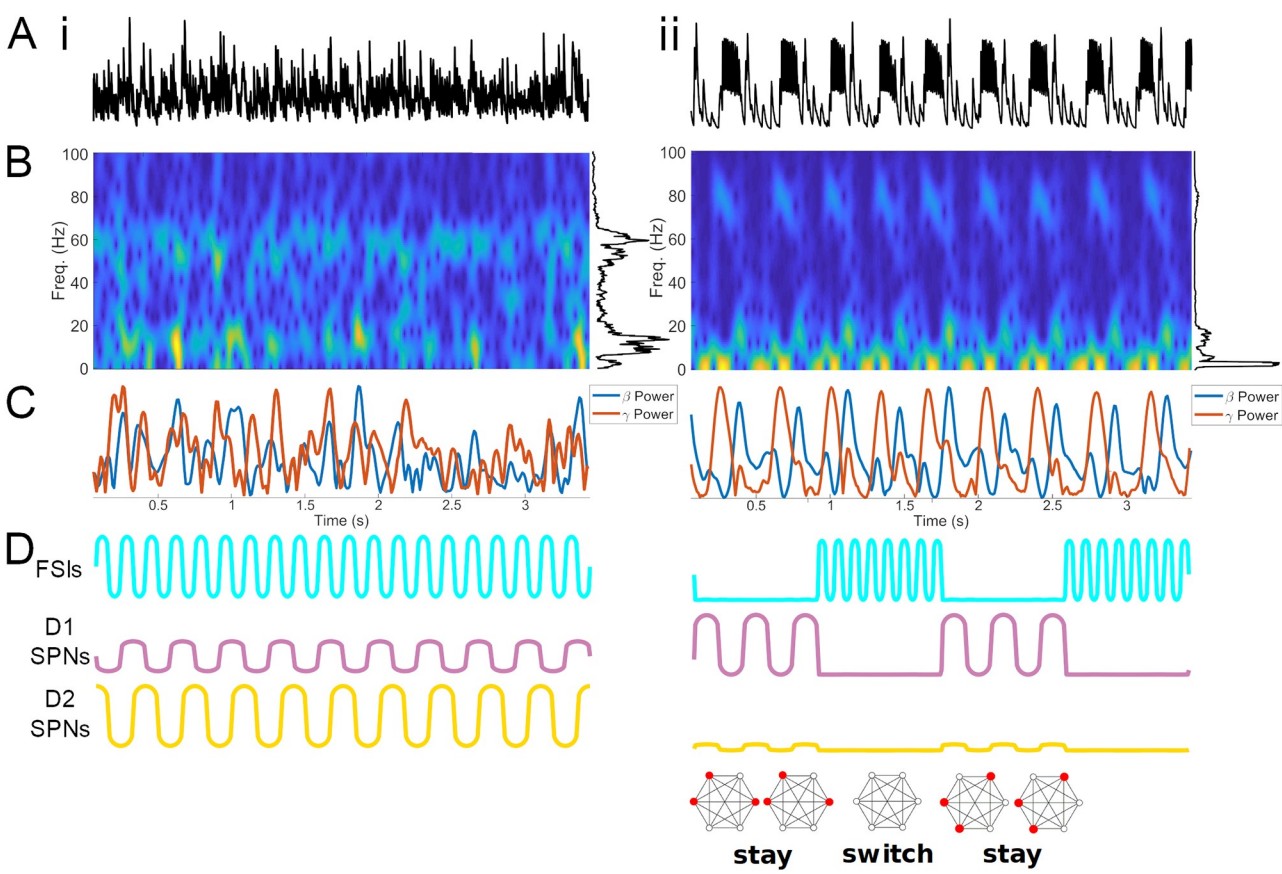

**Fig 7. In the high DA state, packets of FSI *γ* and SPN *β* alternate at a *δ/θ* timescale.** (A) LFP surrogates (summed synaptic currents) for baseline (i) and high (ii) DAergic tone conditions. (B) Spectrograms of LFP surrogates. (C) Wavelet-filtered *β* and *γ* oscillations from the population activity in (A). (D) Schematic of oscillatory activity during baseline and high DAergic tone conditions, with proposed functional impact on ensemble activity.

which produces a low *β* ($\sim$15 Hz). Preliminary data suggest that the SPN network is more sensitive to input in the high DA condition, when the ongoing *β* rhythm is periodically disrupted by the FSI-induced *δ/θ* (S3 Fig).

## Discussion

Our model suggests that DAergic tone can produce a transition between two dynamical states in striatal GABAergic networks. In the baseline DAergic tone state, ongoing low *γ* (55-60 Hz) and *β* ($\sim$15 Hz) oscillations are generated by striatal FSI and SPN networks, respectively (Fig 7i). In the high DAergic tone state, packets of FSI-mediated high *γ* ($\sim$80 Hz) and SPN-mediated *β* (10-20 Hz) rhythms alternate at *δ/θ* ($\sim$3 Hz) frequency (Fig 7ii). Our results make predictions about the generation of striatal rhythms, have implications for the role of FSIs in regulating the activity of SPNs, and suggest an underlying mechanism for the temporal dynamics of motor program selection and maintenance (Fig 7D).

### Mechanisms of *γ* and *δ/θ* oscillations in single FSIs

Prior work has shown *γ* oscillations in striatal FSIs arising from an interaction between the spiking currents and the spike frequency adaptation caused by the potassium D-current,

which produces a minimum FSI firing rate in the $\gamma$ range [47, 61]. The frequency of the FSI $\gamma$ depends on excitatory drive to the FSIs, which in our model leads to the modulation of $\gamma$ frequency by DA, a phenomenon also observed in striatal $\gamma$ oscillations *in vivo* [14, 62–64].

Prior work has also suggested that the D-current is responsible for the bursting or stuttering behavior of FSIs, in which brief periods of high frequency activity are interspersed with periods of quiescence [30]. However, regularity in these periods of quiescence has not been previously observed. Thus, the present study is novel in its prediction of the generation of low-frequency rhythms by FSIs, dependent on high levels of D-current conductance; FSIs have previously been characterized solely as generators of $\gamma$ oscillations. In our model, the D-current is activated by burst spiking, e.g., at $\gamma$ frequency, and hyperpolarizes the cell for roughly a $\delta/\theta$ period due to its long time constant of inactivation. Though the $\delta$ rhythm produced by individual cells decreases in frequency in response to excitatory drive (Fig 1D), the frequency of the resulting $\delta/\theta$ oscillation in the network has a minimum around 3 Hz (Fig 3i). This lower bound on $\delta/\theta$ frequency in the network is likely a result of gap-junction induced synchrony driving burst frequency higher than in the individual cell while maintaining robustness to noise. Notably, this study is also a novel demonstration of the generation of both $\delta/\theta$ and $\gamma$ oscillations by a single membrane current.

## Mechanisms of $\gamma$ and $\delta/\theta$ oscillations in FSI networks

Our model FSI network produces qualitatively different dynamics at high and baseline levels of simulated DA. Under high dopaminergic tone, the FSI network produces high $\gamma$ band (80 Hz) oscillations modulated by a $\delta/\theta$ ($\sim$ 3 Hz) oscillation, while under low dopaminergic tone the FSI network produces low $\gamma$ band (60 Hz) oscillations alone (Fig 4). While both $\delta/\theta$ and $\gamma$ are present at the level of individual cells under all dopaminergic conditions, only in the high DA condition is bursting sufficiently synchronized that $\delta/\theta$ power is present in the network. The presence of $\delta/\theta$ at the network level can be attributed to the higher level of gap junction conductance in the high DA condition (Fig 3Bi).

The ability of gap junctions to generate synchrony is well established in computational and experimental work [34, 41–44, 48, 50, 54, 55, 65, 66]. Previous models from other groups suggest that gap junctions can enable synchronous bursting in interneurons, by aligning the burst envelopes, as in our model [39]. While a shunting effect of low conductance gap junctions can inhibit spiking [54], gap junctions with high enough conductances have an excitatory effect, promoting network synchrony [42, 65]. Previous work has also shown the importance of gap junction connectivity in stabilizing network $\gamma$ oscillations *in silico* [34, 41, 67], as well as network $\gamma$ and $\delta/\theta$ oscillations in inhibitory networks *in vitro* and *in silico* containing noise or heterogeneity [42]. Striatal FSIs *in vivo* are highly connected by gap junctions as well as inhibitory synapses [68], similar to the networks of inhibitory interneurons that produce ING rhythms [40]. Unlike ING, however, our FSI network $\gamma$ is independent of GABAergic synapses: inhibitory conductance has only a small impact on $\gamma$ frequency, and $\gamma$ power is highest when inhibitory synapses are removed (Fig 3C). In slice, the $\gamma$ resonance of striatal FSIs is dependent on gap junctions but not on GABA [69], suggesting that our model is an accurate representation of striatal FSI $\gamma$.

It is important to note that, while our model is conceived as a representation of the striatal microcircuit, physiologically similar FSI networks are present in cortex [30]; therefore, the mechanisms described here may contribute to the generation of $\delta/\theta$-modulated $\gamma$ oscillations in cortex as well.

## Support for striatal rhythm generation

Our model provides mechanistic explanations for all four oscillatory bands observed in ventral striatum *in vivo* ($\delta/\theta$, $\beta$, low $\gamma$, and high $\gamma$) [70]. Previous modeling and experiments suggest $\beta$ can be generated by striatal SPNs [45, 71, 72]. Our results suggest that FSIs generate striatal $\gamma$, and that motor- and reward-related increases in $\gamma$ power reflect increased striatal FSI activity.

There is evidence to support the existence of a locally generated striatal $\gamma$ oscillation that is not volume conducted and that responds to local DAergic tone [27, 73]. The FSIs of the striatum are the most likely candidate generator of this rhythm: they are unique among striatal cell types in preferentially entraining to periodic input (from each other and from cortex) at $\gamma$ frequencies [5, 44, 74–76]. Different populations of striatal FSIs *in vivo* entrain to different $\gamma$ frequencies, and FSIs entrained to higher frequencies are also more entrained to cortical input [14, 62–64, 70]. It is likely that different subpopulations of FSIs selectively entrain to specific $\gamma$ frequencies, determined by physiological and contextual, including neuromodulatory (e.g., DAergic), factors.

Experimental evidence also supports striatal FSI involvement in a DA-modulated $\delta$ or $\theta$ rhythm. FSIs phase lock to spontaneous striatal LFP oscillations at $\delta$ [22, 77, 78] and $\theta$ [60, 79–81] as well as $\gamma$ frequencies. *In vivo*, striatal $\delta$ and $\theta$ power are modulated by task-related phenomena such as choice points and motor execution, as well as by reward and reward expectation, suggesting responsiveness in both frequency bands to DA (known to phasically increase in response to reward cues) [12, 82–86]. $\theta$ has also been shown to modulate the response of SPNs to reward [15].

The slow rhythm generated by our model network is on the boundary between the $\delta$ and $\theta$ frequency bands, and as such it is difficult to determine for which of the two bands our model has more substantial implications. However, many electrophysiological studies of striatum find a low frequency rhythm in this intermediate 3 to 5 Hz range [28, 87, 88]. While rodent electrophysiology suggests that $\delta$ is more prevalent in the striatum of the resting animal and $\theta$ is stronger during high DAergic tone [89, 90], human studies suggest that DAergic reward signals are associated with increased power in the $\delta$ band in nucleus accumbens and that $\theta$ power (which originates in cortex) is associated with the decreased DA signal following an unexpected loss [91, 92]. The frequency of this slow rhythm may be determined by entrainment to rhythmic cortical input, or by different subpopulations of cells responding to different components of the dopamine signal (e.g. tonic versus phasic, anticipatory vs consummatory, etc.).

The $\beta$ oscillations produced by our model network vary in frequency. During simulated baseline DAergic tone, the $\beta$ frequency in both SPN subnetworks is closer to 15 Hz, while during high DAergic tone, the $\beta$ frequency produced by the D1 SPN subnetwork approaches 20 Hz, without a change in the frequency generated by the D2 SPN subnetwork (Figs 5 and 6). This behavior is not unexpected, as our previous modeling work suggested that the frequency of the $\beta$ generated by SPN networks is sensitive to excitatory drive [45], which is the difference between the cell subtypes in this model. Experimental evidence also supports the association of low-$\beta$ but not high-$\beta$ frequencies with the indirect (D2-expressing) pathway of the basal ganglia [93]. Corticostriatal models constructed by our group that include connectivity differences between D1 SPNs and D2 SPNs suggest that these differences in $\beta$ frequency may be an essential component of how cortical input is routed to the direct versus the indirect pathway during decision making [94].

## Rhythmicity in striatal dynamics and movement

*In vivo*, striatal $\beta$ power has a well established negative correlation with DA and locomotion in both health and disease, while striatal $\gamma$ power has a positive correlation with both [2, 11, 12,

19, 20, 95]. $\beta$ oscillations in the basal ganglia are thought to provide a "stay" or "status quo" signal that supports maintenance of the currently active motor program [46], and they are causally implicated in motor slowing and cessation [16, 17, 21, 25, 95, 96].

In our simulations of high DAergic tone, FSI spiking at high $\gamma$ frequencies $\delta/\theta$-periodically inhibits SPN-generated $\beta$ oscillations, permitting SPN $\beta$ only during the 150-200 millisecond $\delta/\theta$ trough corresponding to the FSIs' interburst interval. We hypothesize that these periodic gaps between SPN $\beta$ packets are necessary to terminate ongoing motor programs and initiate new motor programs, both represented by active SPN cell assemblies. During the $\delta/\theta$ trough, all SPN cell assemblies are simultaneously released from inhibition and viable to compete once again to determine the current motor program, with incoming input from cortex influencing this competition. Under this interpretation, our results predict that striatal networks oscillate between a "stay" or "program on" state marked by SPN $\beta$ oscillations, and a "switch" or "program off" state marked by FSI high $\gamma$ oscillations, and that the $\delta/\theta$ period limits the speed of sequential motor program execution (Fig 7D). Accordingly, the SPN network responds more specifically to input when the FSI-induced $\delta/\theta$ is periodically disrupting the intrinsic SPN $\beta$ rhythm (S3 Fig). Associations formed between sets of SPNs receiving similar input persist during an ongoing $\beta$ oscillation, but these associations are broken by FSI-mediated rhythmic inhibition. This inhibitory disruption thereby allows SPNs to flexibly respond to new input, which would otherwise be unable to override the coordinated activity of pre-existing cell assemblies.

In support of this hypothesis, striatal representations of behavioral "syllables" combined to create motor programs are active for a maximum of $\sim 200$ ms [97], and the velocity of continuous motion is modulated intermittently at a $\theta$ frequency ($\sim 6$-9 Hz) [98]. In healthy animals, the duration of $\beta$ bursts has an upper limit of $\sim 120$ ms, about half a $\theta$ cycle [16], in agreement with our prediction that $\beta$ activity is $\delta/\theta$ phase-modulated. Striatal $\gamma$ has also been observed in transient ($\sim 150$ ms) bursts that are associated with the initiation and vigor of movement [18]. Additionally, other biophysically constrained computational models have suggested that SPN assemblies fire in sequential coherent episodes for durations of several hundred milliseconds, on the timescale of one or several $\delta/\theta$ cycles [99]. Overall, evidence supports the hypothesis that $\beta$ and $\gamma$ oscillations in striatum *in vivo*, and therefore the motor states they encode, are activated on $\delta/\theta$-periodic timescales.

Furthermore, $\beta$ and $\gamma$ power are anticorrelated in EEG and corticostriatal LFP [20, 28, 100], in agreement with our model's prediction that these rhythms are coupled to opposite phases of ongoing $\delta/\theta$ rhythms. FSI and SPN firing are inversely correlated *in vivo*, entrained to $\theta$, and they are active during opposite phases of $\theta$, as observed in our model [60, 79, 101–103]. $\delta/\theta$-$\gamma$ cross-frequency coupling is observed in striatum and increases during reward, when DAergic tone is expected to be high [13, 28, 90, 104, 105]. Our model suggests that these cross-frequency relationships occur in part due to FSI inhibition of SPNs. Though FSIs are smaller in number, FSI-SPN synapses have a much stronger effect than SPN-SPN connections, with each FSI inhibiting many SPNs [59, 106].

During baseline DAergic tone in our model, FSIs produce an ongoing low $\gamma$ that does not effectively suppress SPN $\beta$ activity (produced sporadically in both D1 and D2 SPN networks), and thus does not facilitate the switching of the active SPN assembly. Thus, our model suggests that at baseline levels of DA, switching between SPN assembles may be more dependent on cortical inputs or downstream basal ganglia circuit computations. Although the function of FSI low $\gamma$ input in SPN dynamics is unclear, it may facilitate striatal responsivity to cortical low $\gamma$ input, which occurs in an afferent- and task-specific manner [70]. SPNs do not entrain to $\gamma$ in our model, suggesting that $\gamma$ oscillations are not transmitted to downstream basal ganglia structures.

In contrast, both the $\beta$ and $\delta/\theta$ rhythms in our model entrain SPN networks and may be relayed to other basal ganglia structures. Intriguingly, alternation between $\beta$ and $\gamma$ on a $\delta/\theta$ timescale has been observed in the globus pallidus *in vivo*, and DAergic tone modulates these oscillations and their interactions [28, 87]. Thus, the mechanisms proposed in our model may also play a role in the oscillatory dynamics of other basal ganglia structures, through a combination of rhythm propagation and local rhythm generation by similar circuits. Similar pauses in FSI activity, allowing transient SPN disinhibition and production of $\beta$ oscillations, occur in a recent computational model of the striatal-GPe network [52], also based on an earlier model of stuttering FSIs [30]. In contrast to this work, we emphasize the mechanisms producing $\beta$ and the coordination of $\beta$ and $\gamma$ by $\delta/\theta$, not addressed previously [52].

## Implications for disease

In Parkinson's disease, which is characterized by motor deficits and chronic DA depletion, $\beta$ power is correlated with the severity of bradykinesia [2]. Parkinsonian $\beta$ may be generated by striatal D2 SPNs [45, 71, 72]. Parkinsonian conditions also produce high cholinergic tone [107], known to decrease the conductance of GABAergic FSI-SPN synapses [108]. Thus, the failure of the FSI inhibition-mediated motor program switching described above may play a role in the motor deficits observed in Parkinson's: if DA is low, and FSIs are unable to inhibit either D1 or D2 SPNs, $\delta/\theta$ modulation of SPN $\beta$ rhythmicity will be supplanted by ongoing D2 $\beta$ rhythmicity, impairing motor initiation by reducing the possibility of motor program switching in the Parkinsonian striatum. Supporting this hypothesis, the $\beta$ frequency generated by D2 SPNs in our model is substantially lower than that generated by the D1 SPN subnetwork in the high DA condition (Fig 6). Experimental work suggests that parkinsonian $\beta$ is specifically a low ($<$20 Hz) $\beta$, and that treatment by L-DOPA or deep brain stimulation specifically reduces power in the low $\beta$ band without affecting high $\beta$ power [93, 109, 110]. Our model suggests that this distinction in $\beta$ frequency bands is at least in part due to differences in excitatory drive between subtypes of SPNs expressing different DA receptors.

In hyperkinetic motor disorders, $\gamma$ and $\theta$ rhythms are potentiated: mouse models of Huntington's disease (HD) displays unusually high $\delta/\theta$ and $\gamma$ band striatal LFP power [3, 5, 6]; and L-DOPA-induced hyperkinetic dyskinesia is also characterized by increased high $\gamma$ and $\delta/\theta$ power and reduced $\beta$ power in the striatal LFP [1, 22, 23]. As these rhythms are tied to FSI activation in our model, we suggest that hyperkinetic disorders may result from striatal FSI hyperfunction. Consistent with this hypothesis, in HD model animals, FSI to SPN connectivity is increased, and SPNs respond more strongly to FSI stimulation [7]. Computational modeling suggests that FSI-generated $\gamma$ more readily entrains to $\delta$-frequency input during HD [5].

However, hypofunction of striatal FSI networks can also lead to hyperkinetic disorders, including Tourette's syndrome, dystonia, and dyskinesias [1, 4, 8, 9, 111–113]. Dystonia, which as a disorder of involuntary muscle activation is considered hyperkinetic, can also be characterized by rigidity and freezing due to activation of antagonistic muscles. Indeed, dystonia may be the consequence of an increase in SPN firing rate due to D2 receptor dysfunction [114]. Our model suggests that FSI hypofunction may contribute to dystonia by resulting in excessive SPN $\beta$ rhythmicity and decreased probability of motor program switching. A reduction in $\theta$-$\gamma$ cross frequency coupling has been reported in L-DOPA-induced dyskinesia, suggesting that a chronic hyperkinetic high-DA state may also abolish the FSI-generated $\delta/\theta$-coupled $\gamma$ produced here, possibly by pushing the FSI out of its bursting regime and into a tonic spiking mode [24]. These findings underscore the importance of balanced FSI inhibition of SPNs, exemplified by the periodic suppression observed in our model, which we suggest enables the flexible striatal network activity that allows for smooth, purposeful movements.

## Caveats and limitations

Little experimental evidence on the striatal FSI D-current conductance exists. The level of D-current conductance we've chosen leads to $\gamma$ frequencies and FSI firing rates that are more in line with experimental observations than with previous models; this level of D-current also produces $\delta/\theta$ rhythmicity in FSI networks. Our parameter choices result in a model exhibiting a transition between "low DA" and "high DA" dynamic states that matches experimental observations and has powerful functional interpretations. Validating our results will require further experimental investigation of the D-current in striatal FSIs. Interestingly, DA has been shown to downregulate D-current conductance in prefrontal cortical FSIs [115]. If striatal FSIs exhibited a similar DA-dependent D-current downregulation, our simulations suggest that the transition between high and low DA states could be different from that described in the current study. The existence and functional interpretations of other dynamic transitions are beyond the scope of this paper.

In general, many DA-dependent changes in striatal neurophysiology have been observed. For the sake of simplicity, most of these have been left out of our modeling. For example, D1 and D2 SPNs respond differently to adenosine [116] and peptide release [117], but we did not consider these significant factors in the production of striatal $\beta$ oscillations. While the nature of the changes induced by DA in our network is based on a review of the literature, the actual values chosen are assumptions of the model. Details on the rationale behind each specific value are given in the Methods section.

We also omitted inhibitory connections between D1 and D2 SPN populations. The connectivity from D1 to D2 SPNs is very sparse (6 percent). Connections from D2 to D1 SPNs are more prevalent, but it seems unlikely that these projections would qualitatively alter our results: during the baseline state, the D1 and D2 SPNs are identical; during the high DA state, SPN inhibition tends to increase SPN $\beta$ rhythmicity and spiking.

In our model the number of FSIs is small, so every FSI participates on every $\theta$ cycle; *in vivo*, the participation of multiple FSI populations is likely coordinated by cortex. Coordinated FSI activity has proven hard to observe over long periods *in vivo* [14, 118]. However, FSIs form local functional circuits [119], and *in vivo*, striatal FSI assemblies exhibit transient gap-junction dependent synchronization [66], possibly resulting from brief bouts of correlated cortical or homogeneous DAergic input. Furthermore, different subpopulations of FSIs have strong preferences for projecting to either D1 or D2 SPNs, as opposed to the overlapping projections modeled in our current study, and these distinct populations respond differently to cortical oscillations [80]. Thus, local $\gamma$ synchrony may exist in small striatal subnetworks and be amplified by DA or cortical input via the differential recruitment of multiple FSI subpopulations.

Compounding the issues of unrealistic population size, the ratio of FSIs to SPNs in our model is much higher than data from rodent striatum suggest. 20% of the cells in our model network are FSIs, while FSIs comprise only about 0.7-1% of cells in rodent striatum [120]. Unfortunately, it would be computationally intractable to reproduce the network dynamics of the present model at a ratio of 50 or 100 SPNs per FSI. However, in humans the proportional number of FSIs is much higher; interneurons may account for as many as 25% of human striatal neurons [121]. We have attempted to structure our model such that each SPN receives a realistic number of incoming connections from FSIs (mean 18.75 in our model, based on a range of 4 to 27 [29]), and such that these synapses are of realistic strengths. Therefore, it is reasonable to predict that the qualitative dynamics of FSI to SPN inhibition in our model would be similar even if the number of SPNs present were much higher.

Finally, cortical input to both FSIs and SPNs was simulated as Poisson noise. In a sense, we simulated a model of striatum to which cortex is not providing informative input. It could be

the case that this is a population that is not "selected" by cortex to take part in motor activity, a population that is in a "listening" state awaiting cortical input, or a population taking part in a learned behavior that can be executed without cortical input. However, cortical input is probably essential in determining which SPNs and FSIs take part in network oscillatory activity. If the FSIs play a role in organizing the response of the SPNs to cortical input, changing the properties of the simulated input may prove informative in terms of how this organization might take place. In particular, cortical inputs may be more correlated within certain FSI subpopulations than others. Previous modeling work has shown that networks of striatal FSIs can detect correlated input [54], a property that may play an important computational role in striatal function. Additionally, we can expect that input from cortex has oscillatory properties of its own. Exploring these complexities is an important direction for future research into the role of striatal GABAergic networks and rhythmic dynamics in motor behavior.

## Materials and methods

All neurons (FSIs and SPNs) are modeled using conductance-based models with Hodgkin-Huxley-type dynamics. SPNs are modeled with a single compartment and FSIs have two compartments to represent the soma and a dendrite. The temporal voltage change of each neuron is described by (Eq 1):

$$c_m \frac{dV}{dt} = -\sum I_{memb} - \sum I_{syn} + I_{\text{app}} \qquad (1)$$

Membrane voltage ($V$) has units of $mV$. Currents have units of $\mu A/cm^2$. The specific membrane capacitance ($c_m$) is 1 $mF/cm^2$ for all FSIs and SPNs. Each model neuron has intrinsic membrane currents ($I_{memb}$) and networks of neurons include synaptic currents ($I_{syn}$). The applied current term ($I_{\text{app}}$) represents background excitation to an individual neuron and is the sum of a constant and a noise term.

All membrane currents have Hodgkin-Huxley-type conductances formulated as:

$$I = \bar{g}(m^n h^k)(V - E_{ion}) \qquad (2)$$

Each current in Eq 2 has a constant maximal conductance ($\bar{g}$) and a constant reversal potential ($E_{ion}$). The activation ($m$) and inactivation ($h$) gating variables have $n^{th}$ and $k^{th}$ order kinetics, where $n, k \geq 0$. The dynamics of each gating variable evolves according to the kinetic equation (written here for the gating variable $m$):

$$\frac{dm}{dt} = \frac{m_\infty - m}{\tau_m} \qquad (3)$$

The steady-state functions ($m_\infty$) and the time constant of decay ($\tau_m$) can be formulated using the rate functions for opening ($\alpha_m$) and closing ($\beta_m$) of the ionic channel by using the equations:

$$m_\infty = \alpha_m/(\alpha_m + \beta_m)$$
$$\tau_m = 1/(\alpha_m + \beta_m).$$

The specific functions and constants for different cell types are given below.

### Striatal fast spiking interneurons

Striatal fast spiking interneurons (FSIs) were modeled as in Golomb et al., 2007 [30], using two compartments. The voltage in the somatic compartment ($V$) and in the dendrite ($V_d$) evolve

according to:

$$c_m \frac{dV}{dt} = -I_{Na} - I_K - I_L - I_D - I_{syn} + I_{ds} \qquad (4)$$

$$c_m \frac{dV_d}{dt} = -I_{Na} - I_K - I_L - I_D - I_{syn} + I_{ext} + I_{sd} \qquad (5)$$

Background excitation is represented by the term $I_{ext}$, which is formulated as the sum of a tonic, DA dependent current and Poisson input. The units of $I_{ext}$ are in $\mu A/cm^2$. The tonic, DA dependent current is discussed below. Each FSI receives independent, excitatory Poisson input with a rate of 100 inputs per second.

The synaptic current ($I_{syn}$) is the sum of GABA$_A$ currents and electrical connections between FSIs (formulated below). The FSI membrane currents ($I_{memb}$) consisted of a fast sodium current ($I_{Na}$), a fast potassium current ($I_k$), a leak current ($I_L$), and a potassium D-current ($I_D$). The formulations of these currents were taken from previous models of striatal FSIs [30, 47]. $I_{ds}$ represents the current from the dendritic compartment to the somatic compartment and $I_{sd}$ represents the current from the somatic compartment to the dendritic compartment.

The maximal sodium conductance is $\bar{g}_{Na} = 112.5 \; mS/cm^2$ and the sodium reversal potential is $E_{Na} = 50$ mV. The sodium current has three activation gates ($n = 3$) and one inactivation gate ($k = 1$). The steady state functions for the sodium current activation ($m$) and inactivation ($h$) variables and their time constants ($\tau_m$ and $\tau_h$, respectively) are described by:

$$m_\infty = \frac{1}{1 + \exp\left[-(V + 24)/11.5\right]} \qquad (6)$$

$$h_\infty = \frac{1}{1 + \exp\left[(V + 58.3)/6.7\right]} \qquad (7)$$

$$\tau_h = 0.5 + \frac{14}{1 + \exp\left[(V + 60)/12\right]} \qquad (8)$$

The maximal conductance for the fast potassium channel is $\bar{g}_K = 225 \; mS/cm^2$ and the reversal potential for potassium is $E_K = -90$ mV. The fast potassium channel has no ($k = 0$) inactivation gates but has two ($n = 2$) activation gates described by its steady state function ($n_\infty$) and time constant ($\tau_n$):

$$n_\infty = \frac{1}{1 + \exp\left[-(V + 12.4)/6.8\right]} \qquad (9)$$

$$\tau_n = \left(0.087 + \frac{11.4}{1 + \exp\left[(V + 14.6)/8.6\right]}\right)\left(0.087 + \frac{11.4}{1 + \exp\left[-(V - 1.3)/18.7\right]}\right) \qquad (10)$$

The leak current ($I_L$) has no gating variables ($n = 0, k = 0$). The maximal leak channel conductance is $g_L = 0.25 \; mS/cm^2$ and the leak channel reversal potential is $E_L = -70$ mV.

The fast-activating, slowly inactivating potassium D-current ($I_D$) is described mathematically as in Golomb et al, 2007 [30] and has three activation gates ($n = 3$) and one inactivation ($k = 1$) gate. The steady state functions for the activation ($a$) and inactivation ($b$) variables are

formulated as:

$$a_\infty = \frac{1}{1 + \exp\left[-(V + 50)/20\right]} \tag{11}$$

$$b_\infty = \frac{1}{1 + \exp\left[(V + 70)/6\right]} \tag{12}$$

The time constant of the decay is 2 ms ($\tau_a$) for the activation gate and 150 ms ($\tau_b$) for the inactivation gate. The maximal conductance of the D-current is 6 $mS/cm^2$.

All conductances in the dendritic compartment of the FSIs ($g_{Na}$, $g_K$, $g_D$, $g_L$) are 1/10 the strength of those in the somatic compartment. The somatic and dendritic compartment of each cell are connected bidirectionally with a compartmental conductance of 0.5 $mS/cm^2$. This electrical coupling is formulated as:

$$I_{sd} = 0.5(V_{soma} - V_{dend}) \tag{13}$$

$$I_{ds} = 0.5(V_{dend} - V_{soma}) \tag{14}$$

where $I_{sd}$ is the current from the somatic compartment to the dendritic compartment and $I_{ds}$ is the current from the dendritic compartment to the somatic compartment.

## Striatal spiny projection neurons

Spiny projection neurons were modeled with four membrane currents: a fast sodium current ($I_{Na}$), a fast potassium current ($I_k$), a leak current ($I_L$), and an M-current ($I_m$) [31]. We do not model SPN up and down states which are not prevalent in the awake state of striatum [122], the state being modeled, and therefore we do not include the Kir current in our model, which is active during the SPN down state.

The sum of all excitatory inputs from the cortex and thalamus and inhibitory inputs from striatal interneurons is introduced into the model using a background excitation term ($I_{app}$). $I_{app}$ is the sum of a constant term and a Gaussian noise term. The Gaussian noise has mean zero, standard deviation one and an amplitude of $4\sqrt{\delta t}$ where $\delta t$ is the time step of integration. D1 and D2 SPNs were distinguished only by the value of tonic term of $I_{app}$ when DA levels were high. DA is excitatory to D1 receptors and inhibitory to D2 receptors [123]. Thus, we modeled D1 and D2 SPNs as having the same tonic $I_{app}$ at baseline DAergic tone state with $I_{app} = 1.19 \ \mu A/cm^2$. To model the high DA state, let the tonic term of $I_{app} = 1.29 \ \mu A/cm^2$ for the D1 SPNs and $I_{app} = 1.09 \ \mu A/cm^2$ for the D2 SPNs.

The rate functions for the sodium current activation ($m$) and inactivation ($h$) variables are formulated as:

$$\alpha_m = \frac{0.32(V + 54)}{1 - \exp\left[-(V + 54)/4\right]} \tag{15}$$

$$\beta_m = \frac{0.28(V + 27)}{\exp\left[(V + 27)/5\right] - 1} \tag{16}$$

$$\alpha_h = 0.128 \exp\left[-(V + 50)/18\right] \tag{17}$$

$$\beta_h = \frac{4}{1 + \exp\left[-(V + 27)/5\right]} \tag{18}$$

The maximal conductance of the sodium current is $\bar{g}_{Na} = 100 \ mS/cm^2$. The sodium reversal potential is $E_{Na} = 50$ mV. The sodium current has three activation gates ($n = 3$) and only one inactivation gate ($k = 1$).

The fast potassium current ($I_K$) has four activation gates ($n = 4$) and no inactivation gates ($k = 0$). The rate functions of the activation gate are described by:

$$\alpha_m = \frac{0.032(V + 52)}{1 - \exp\left[-(V + 52)/5\right]} \tag{19}$$

$$\beta_m = 0.5 \exp\left[-(V + 57)/40\right] \tag{20}$$

The maximal fast potassium channel conductance is $\bar{g}_K = 80 \ mS/cm^2$. The reversal potential for potassium is $E_K = -100$ mV.

The leak current ($I_L$) has no gating variables ($n = 0, k = 0$). The maximal conductance of the leak channel is $g_L = 0.1 \ mS/cm^2$. The leak channel reversal potential is $E_L = -67$ mV.

The M-current has one activation gate ($n = 1$) and no inactivation gate ($k = 0$). The rate functions for the M-current activation gate are described by:

$$\alpha_m = \frac{Q_s 10^{-4}(V + 30)}{1 - \exp\left[-(V + 30)/9\right]} \tag{21}$$

$$\beta_m = -\frac{Q_s 10^{-4}(V + 30)}{1 - \exp\left[(V + 30)/9\right]} \tag{22}$$

We use a $Q_{10}$ factor of 2.3 to scale the rate functions of the M-current since the original formulation of these kinetics described dynamics at 23°C [124]. Thus, for a normal body temperature of 37°C, the M-current rate equations are scaled by $Q_s$, which is formulated as:

$$Q_s = Q_{10}^{(37 \,°C - 23 \,°C)/10} = 3.209 \tag{23}$$

The maximal M-current conductance is $\bar{g}_m = 1.25 \ mS/cm^2$.

## Synaptic connectivity and networks

Networks of FSIs contained 50 neurons. For networks that additionally had SPNs, we modeled 100 D1 SPNs and 100 D2 SPNs. The model synaptic GABA$_A$ current ($I_{GABA_A}$) is formulated as in McCarthy et al., 2011 [45] and is the only synaptic connection between SPNs and from FSIs to SPNs. The GABA$_A$ current has a single activation gate dependent on the pre-synaptic voltage.

$$I_{GABA_A} = \bar{g}_{ii} s_i (V - E_i) \tag{24}$$

The maximal GABA$_A$ conductance between FSIs is $\bar{g}_{ii} = 0.1 \ mS/cm^2$. Conductances from FSIs to SPNs and between SPNs (but not between FSIs) were normalized to the number of SPNs in the target network. Therefore, the maximal GABA$_A$ conductance from FSIs to SPNs is $\bar{g}_{ii} = 0.6/100 = 0.006 \ mS/cm^2$ and between SPNs was $\bar{g}_{ii} = 0.1/100 = 0.001 \ mS/cm^2$. These values are consistent with FSI to SPN inhibition being approximately six times stronger than inhibition between SPNs [29].

The gating variable for inhibitory GABA$_A$ synaptic transmission is represented by $s_i$. For the $j^{th}$ neuron (FSI or SPN) in the network:

$$s_j = \sum_{k=1}^{N} S_{i_k i_j} \quad (25)$$

The variable $S_{i_k i_j}$ describes the kinetics of the gating variable from the $k^{th}$ pre-synaptic neuron to the $j^{th}$ post-synaptic neuron. This variable evolves in time according to:

$$\frac{dS_{i_k i_j}}{dt} = g_{\text{GABA}_A}(V_k)(1 - S_{i_k i_j}) - \frac{S_{i_k i_j}}{\tau_i} \quad (26)$$

The GABA$_A$ time constant of decay ($\tau_i$) is set to 13 ms for SPN to SPN connections [123] as well as for FSI to FSI connections and FSI to SPN connections [54] The GABA$_A$ current reversal potential ($E_i$) for both FSIs and SPNs is set to -80 mV. The rate functions for the open state of the GABA$_A$ receptor ($g_{\text{GABA}_A}(V_k)$) for SPN to SPN transmission is described by:

$$g_{\text{GABA}_A}(V_k) = 2(1 + tanh(\frac{V_k}{4})) \quad (27)$$

The rate functions for the open state of the GABA$_A$ receptor ($g_{\text{GABA}_A}(V_k)$) for FSI to FSI and FSI to SPN transmission is:

$$g_{\text{GABA}_A}(V_k) = \frac{1}{\tau_r}(1 + tanh(\frac{V_k}{10})) \quad (28)$$

The value of $\tau_r$ is 0.25 ms. FSIs were additionally connected by dendritic electrical connections. The electrical coupling for dendritic compartment i is denoted as $I_{elec}$, has units in $\mu A/cm^2$ and is formulated as:

$$I_{elec} = g_{\text{GJ}}(Vd_j - Vd_i) \quad (29)$$

The value of the gap junction conductance $g_{\text{GJ}}$ depended on DA level (see below). Within the 50-cell FSI network, each pair of FSIs had an independent 33 percent chance of a dendro-dendritic gap junction chosen from a uniform random distribution [54], and an independent 58 percent chance of a somato-somatic inhibitory synapse also chosen from a uniform distribution [53]. SPNs are connected with each other in a mutually inhibitory GABAergic network [125]. We modeled all to all connectivity of inhibitory synapses from any SPN to any SPN of the same receptor subtype, as in [45]. Probability of connection from any given FSI to any given MSN was 37.5 percent, chosen from a uniform random distribution [52, 53].

## Dopamine

DA impacts both connectivity and excitability in the model networks. DAergic tone was simulated as having five components: direct excitation of FSIs [32], increased gap junction conductance between FSIs [33], decreased inhibitory conductance between FSIs [32], increased excitation to D1 SPNs, and decreased excitation to D2 SPNs. DA-induced changes to SPN excitation were discussed above. Excitation to FSIs was modeled as the sum of a tonic, DA dependent input current ($I_{app}$) and a noise term. DA did not change the noise term in either SPNs or FSIs. The baseline DAergic tone state was modeled in FSIs using $I_{app} = 7 \mu A/cm^2$, $g_{GJ} = 0.15 mS/cm^2$ and the GABA$_A$ conductance between FSIs was $g_{ii} = 0.1 mS/cm^2$. The high DA state was modeled in FSIs using $I_{app} = 14 \mu A/cm^2$, $g_{GJ} = 0.3 mS/cm^2$ and $g_{ii} = 0.005 mS/cm^2$. The synaptic conductances were chosen so as to be within an order of magnitude of

physiological estimates (0.05 $mS/cm^2$ for $g_{GABA_A}$ [52, 53]; 0.2 $mS/cm^2$ for $g_{GJ}$ [126]). The inhibitory conductance for the high DAergic tone state was chosen to be the lowest value possible in this range; the inhibitory conductance for the low DAergic tone state was chosen to be the highest value that would still reliably allow oscillatory behavior in the network. The value of $g_{GJ}$ in the low DAergic condition was then chosen to be the lowest value that was permissive of oscillatory behavior, and the value in the high DAergic condition was chosen to be twice that. Finally, the values of $I_{app}$ were chosen in order to correspond to physiologically realistic firing rates (a minimum of 5 and a maximum of 30 Hz; see [103, 118]).

### Local field potential

The local field potential (LFP) was calculated as the sum of all synaptic currents in all cells. Stationarity of the network appears in the raster plots after about 500 ms. To eliminate transients due to initial conditions, our LFP is evaluated only after 1,000 ms of simulated time. We estimated the power spectral density of the simulated LFP using the multitaper method. [127].

### Simulations

All simulations were run on the MATLAB-based programming platform DynaSim, a framework for efficiently developing, running and analyzing large systems of coupled ordinary differential equations, and evaluating their dynamics over large regions of parameter space [128]. DynaSim is open-source and all models have been made publicly available using this platform. All differential equations were integrated using a fourth-order Runge-Kutta algorithm with time step .01 ms. Plotting and analysis were performed with inbuilt and custom MATLAB (version 2017b) code.

### Supporting information

**S1 Fig. Low frequency oscillations are more robust to noise in the high dopamine FSI network than in a single FSI.** (A) Plot of normalized low frequency (<10 Hz) power of the voltage of a single model FSI (blue) and the summed voltages of the high DA FSI network (red) as Poisson noise of varying rate is applied. Each cell in the network receives the same amount of noise that the isolated cell receives. $I_{app}$ = 14 $\mu A/cm^2$ for all simulations; in the high DA FSI network, $g_{gap}$ = 0.3 $mS/cm^2$, $g_{syn}$ = 0.005 $mS/cm^2$. The solid line represents the mean value over 10 simulations per point. Shading represents standard deviation from these means. Power spectra are derived using Thomson's multitaper power spectral density (PSD) estimate (MATLAB function pmtm). (B) Plot of normalized low frequency (<10 Hz) power of the voltage of a single model FSI and the summed voltages of the high DA FSI network as Poisson noise of varying amplitude is applied.
(TIF)

**S2 Fig. FSI network rhythms are robust to noise and heterogeneity.** Power and frequency of $\delta/\theta$ and $\gamma$ rhythms in FSI network mean voltage as a function of (A) noise frequency, (B) noise amplitude, (C) heterogeneity in leak current conductance, (D) heterogeneity in potassium D current conductance, and (E) heterogeneity in applied current. For heterogeneity values, 0 represents completely uniform values and 1 represents a level of heterogeneity where values vary between zero and twice the default value. Default leak current conductance is 0.25 $mS/cm^2$ and default D current conductance is 6 $mS/cm^2$; default applied current is 7 $mA/cm^2$ for low DA and 14 $mA/cm^2$ for high DA. The parameters not being varied in plots A-C are held at either the high DA values (solid lines, $I_{app}$ = 14 $\mu A/cm^2$, $g_{gap}$ = 0.3 $mS/cm^2$, $g_{syn}$ = 0.005 $mS/cm^2$) or the low DA values (dotted lines, $I_{app}$ = 7 $\mu A/cm^2$, $g_{gap}$ = 0.15 $mS/cm^2$, $g_{syn}$ = 0.1 $mS/cm^2$),

according to the legend. The solid line represents the mean value over 10 simulations per point. Shading represents standard deviation from these means. Power spectra are derived using Thomson's multitaper power spectral density (PSD) estimate (MATLAB function pmtm).
(TIF)

**S3 Fig. SPN assemblies are more readily formed in response to new input when FSIs are imposing a $\delta/\theta$ rhythm that disrupts prior activity.** (A) Example raster plot of the D1 SPN subnetwork receiving $\delta/\theta$ frequency FSI input while being subjected to input during high DAergic tone: An excitatory 20 millisecond pulse of input is provided to cells 50-100 (assembly 1) at t = 1680 ms and a later excitatory pulse of input is provided to cells 25-75 (assembly 2) at t = 2080 ms. Assembly 1 is active for several $\beta$ cycles after the first input, causing rebound spiking at antiphase of the cells not in assembly 1 (as in McCarthy 2011 [45]), but becomes inactive during the $\delta/\theta$ peak beginning around t = 1800 ms. Assembly 2 can then respond with a high degree of coherence shortly after the second input. (B) Example raster plot of the isolated D1 SPN subnetwork (not receiving any FSI input) being subjected to the input during high DAergic tone. The same two excitatory pulses are provided. Assembly 1 and its antiphase activity begin firing similarly to the example in (A), but since there is no $\delta/\theta$ input, the $\beta$-rhythm firing of assembly 1 persists indefinitely. Input to assembly 2 is thereby unable to generate a specific response, and the coherence of assembly 1 persists even after the second input. (C) Plot showing history-independence of SPN responses when FSIs are present. Regardless of the phase at which input is given, the maximal response of SPNs in any given cell assembly occurs at a preferred $\delta/\theta$ phase around -2 radians, "erasing" the information of when the input arrived. When FSIs are not present, there is no theta rhythm in the network, and the response of the cells to input is more random.
(TIF)

**S1 File. Complete DynaSim code for reproduction of figures in this manuscript.**
(ZIP)

## Acknowledgments

Jason Sherfey and Erik Roberts developed the DynaSim Toolbox used to run the simulations in this manuscript and assisted with debugging and visualization. We thank Austin Soplata and Amelie Aussel for testing the publically available code and providing input on S3 Fig. In addition, we thank Sean Patrick for assistance in interpreting the behavioral implications of our model.

## Author Contributions

**Conceptualization:** Julia A. K. Chartove, Michelle M. McCarthy, Benjamin R. Pittman-Polletta, Nancy J. Kopell.

**Formal analysis:** Julia A. K. Chartove.

**Funding acquisition:** Nancy J. Kopell.

**Investigation:** Julia A. K. Chartove.

**Methodology:** Julia A. K. Chartove, Michelle M. McCarthy, Benjamin R. Pittman-Polletta, Nancy J. Kopell.

**Project administration:** Benjamin R. Pittman-Polletta, Nancy J. Kopell.

**Software:** Julia A. K. Chartove, Benjamin R. Pittman-Polletta.

**Supervision:** Michelle M. McCarthy, Benjamin R. Pittman-Polletta, Nancy J. Kopell.

**Visualization:** Julia A. K. Chartove, Benjamin R. Pittman-Polletta.

**Writing – original draft:** Julia A. K. Chartove, Michelle M. McCarthy, Benjamin R. Pittman-Polletta, Nancy J. Kopell.

**Writing – review & editing:** Julia A. K. Chartove, Michelle M. McCarthy, Benjamin R. Pittman-Polletta, Nancy J. Kopell.

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
