## [Decision Letter · Decision Letter 0]

1 Sep 2019

Dear Dr Chartove,

Thank you very much for submitting your manuscript 'A biophysical model of striatal microcircuits suggests theta-rhythmically interleaved gamma and beta oscillations mediate periodicity in motor control' for review by PLOS Computational Biology. Your manuscript has been fully evaluated by the PLOS Computational Biology editorial team and in this case also by independent peer reviewers. The reviewers consider in general the manuscript well written and scientifically sond but raised some concerns. While your manuscript cannot be accepted in its present form, we are willing to consider a revised version in which the issues raised by the reviewers have been fully addressed. Please note while forming your response, if your article is accepted, you may have the opportunity to make the peer review history publicly available. The record will include editor decision letters (with reviews) and your responses to reviewer comments. If eligible, we will contact you to opt in or out.

Sincerely,

Michele Migliore

Guest Editor

PLOS Computational Biology

Lyle Graham

Deputy Editor

PLOS Computational Biology

[LINK]

Reviewer's Responses to Questions

**Comments to the Authors:**

Reviewer #1: General comment: the Authors propose a novel model of striatal microcircuitry that explains how fast-spiking interneurons (FSIs) are key in allowing flexibility in motor programs, by modulating the dynamic oscillatory behavior in the network. While this work is relevant for the research in the field and generates interesting predictions, the results and, more importantly, the figures could be made more readable. Some examples are provided below.

On the other hand, the Authors made a great effort in discussing their findings, interpreting them in the context of motor behavior and diseases and highlighting the limitations of the model. One of the main limitation is the choice of the conductance value of the D-current for which, as the Authors acknowledge, there is little experimental evidence. The choice of this value is key for the results presented here.

Data availability: the Authors made the source code of their models publicly available on Github (https://github.com/jchartove/striatum), in agreement with the Journal policies, but no explanation is provided. In order to make the findings easily reproducible, it is of major importance that the source code is accompanied by an explanation on how to run the different scripts, which figure(s) could be easily reproduced, etc... (= add a readme file and description).

The abstract (340 words) exceeds the maximum number of words (which is 300). The Authors could, for instance, move the detailed explanation of the model to the introduction.

Lines 12-13: delta oscillations could be mentioned here, because they appear later in the next without previous introduction (line 77).

Could the Authors better explain why dopamin (DA) has a direct excitatory effect on FSIs? Is this supported experimentally? This effect is also important for some of findings: e.g. DA causes the switch of oscillatory activity in FSIs. Is it a model hypothesis? They cite [12] (Methods, line 550; line 33 and 38), however [12] is about modulation of gamma and beta oscillations in hippocampus and there's no mention of neither dopamin nor the striatum.

Figure 2 caption: "I_D" has not been introduced before

Figure 2:

- Why the orange line starts only at x = 6 and not at x=0?

- In C the axis labels and the legend are a bit difficult interpret: blue line is the frequency of theta oscillations and orange is the power according to the legend, why is there something different written on the axis label?

Could the authors explain how was the connection probability between FSIs chosen? If there is no experimental data supporting the value of 0.3, it should mentioned among the hypothesis of the model or potential limitations.

Figure 4:

- It would be useful to have a legend in panel A. It could be somewhat understood that the arrow represents DA, etc...but a legend would make it clearer.

- Also panel D would benefit from a legend. What is the shade? Is it the standard deviation of the average power spectrum of the cells? Why there is no shade in Fig4 Dii? Is this explained at lines 137-138?

Lines 128-130: this is probably more clear in Fig. 4Di, not 4Ei

The Authors states that DA increases the tonic excitation in the spiny projection neuron, type D1 (SPNs D1), while it decreases tonic excitation in SPNs D2. Why in Figs. 5Ai and 5Aii, 6Ai and6Aii the arrows to SPNs are inverted in the low and high DA conditions? As above, a legend would help.

Line 179: this sentence is incomplete.

The Authors found that in the network model with SPNs and FSIs, FSIs generate low gamma oscillations (line 184). This result is supported by Fig.1, Fig.2, Fig.4. However these figures refer to models with only FSIs. In Fig. 5iii and 5iv, where the full network (FSIs and SPNs) is shown, there is no clear evidence of low gamma oscillations. The only evidence is after applying wavelet-filtering to the LFP surrogate (Fig.6iD). Could the Authors explain in more detail the analysis? The data shown in Fig.6 seem to be the average of the whole network activity. Do the Authors think that the behavior of isolated FSIs networks are maintained in the full network with SPNs?

Line 320: "BG" acronym has not been introduced before.

Line 364: "regieme" should be "regime".

Reviewer #2: The Authors present the analysis of the results obtained simulating biophysical models of the striatal single neurons and microcircuit. In particular, they have observed the evolution of oscillation frequencies of the generated Local Field Potential (when analyzing a population) or membrane voltage (when analyzing a single neuron), at different frequency bands (delta, theta, beta, low/high gamma).

This work focused on the effect of Dopamine levels on the frequency behaviour of the striatal microcircuit; its main contribution is the mechanistic explanation, by means of biophysical neural models, of the theta-modulation of striatal activity caused by high levels of Dopamine. This can justify the observed misbehaviours in motor program execution in pathological conditions, where Dopamine is impacted (e.g., Parkinson's Disease)

The modelling work is sound and well explained, the Authors have organized the manuscript in a way that guides the reader from the single neuron dynamics to the network-level dynamics, explaining at each step the causal-effect mechanisms. The claims are supported by the data presented and are novel, at the same time founded on previous experimental and computational studies on basal ganglia. Relevant literature has been appropriately included in the construction of the model and in the discussion of the results.

In addition to some minor comments, reported below, I just have three concerns:

1) The model of the FSI - D1 SPN - D2 SPN striatal network has been built using 100 neurons per population. However, SPNs make up 95% of striatal neurons in rodents, while interneurons consist of at most 5% of striatal cells. The different ratios in the number of neurons used to build the model and the ones found in the biological network are very different. The Authors addresses this point only in the Methods section (from Line 511). However, this evident discrepancy, given its importance, should be highlighted in the "Caveats and Limitations" section. The Authors should discuss what differences do they expect in the network oscillations with a more biologically-realistic ratio between the number of FSIs and SPNs.

2) The figures could be improved and reorganized in order to facilitate the reader. In the manuscript body, figures are referenced not sequentially and, in some cases, the information regarding the same statement is scattered between different figures (e.g., Line 71: Fig. 1B & 2C; Lines 111-112 Fig. Aii referenced before Fig.Ai; Line 114: Fig. 3Cii referenced before Fig.3B; etc.). It would also be useful to split very large graphics like Fig. 5 and Fig. 6 into smaller and more manageable pieces of information, thus increasing the font size of legends and captions, to improve their readability. Other comments on specific figure improvements are reported below.

3) The Authors have provided a GitHub repository with the codes necessary to run simulations with DynaSim and to reproduce the figures. However, the repository has to be better curated, since it is missing instructions to run the simulations and other details (e.g., what are the versions of MATLAB and the toolbox used). Besides, the codes cannot run since there are paths specified for the developer's machine). I suggest to: i) improve the usability of the provided material ii) use a persistent repository (ModelDB, Harvard Dataverse, etc.) instead of GitHub, to guarantee the preservation and accessibility of the files in the future.

The Data Availability statement should be updated accordingly. Currently, it refers to "Supporting Information files" that do not exist and " Specific data files used to generate the instances of the simulations used for the figures in the manuscript are available upon request" that would not be necessary.

Apart from these three issues, I think the study could make an excellent contribution to the literature, and it fits well within the scope of PLOS Computational Biology, providing sound results to both computational and experimental neuroscience communities studying basal ganglia.

Minor issues:

- In general, please use the conventional space before the unity of measurements in the text (e.g., Fig. 3 Legend, Lines 110, 452, 458, 459, etc.)

- Abstract. Avoid acronyms if possible, to improve its readability.

- Line 13. Define also the delta band ranges.

- Figure 1. Additional titles to Panel Ai and Bii could help. (with I_app = X uA/cm2, ... with Poisson noise lambda = Y, etc.). Legends of Aii and Bii are missing the unit of measurement. Please include I_app and g_d in the labels of C and D.

- Figure 1 Legend. For Bii, please specify what do solid lines and shading represent (median and interquartile ranges?) and why this is the only graphics where multiple measurements are presented.

- Line 74. uA/cm2 is not formatted with mathematical notation

- Line 86. "the model has no minimum firing rate", this seems to be valid only for I_aa <= 3 uA/cm2. The Authors should also explain why the plot with gd=6 stops at I_app = 3 uA/cm2.

- Lines 104, 162, 175, 176. The Authors should motivate their choices for the connectivity probabilities. Are there physiology references that they have used to set those values?

- Line 126. The computation of LFP as the mean voltage of the FSI network is mentioned here for the first time, while it should be introduced at the beginning of the FSI network section.

- Figure 3. I suggest inverting panels i and ii, following the logic flow of the manuscript text. This figure has some visualization problems, for example, the blue shading in Aii, B, ii, Ci, and Cii is partially hiding the mean/median blue line.

- Figure 3 Legend. Please specify what do solid lines and shading represent.

- Figure 4. Panel A. The symbols used for GABA conductance, external excitation, and gap junction conductance should be explained (at least in the legend). The lower-panels E should be clearly marked as insets of higher-panels E.

- Figure 4 Legend. The Authors should clarify the difference between "FSI population activity" and "all individual voltage traces in the network". Is one of the two the LFP?

- Line 155. "This previous work", it is unclear what is the reference.

- Figure 5. As in Fig. 4, symbols in Panels A should be explained. Panels B are missing indications of the x-axis and y-axis scales and labels. The Authors should add to Panels that the higher and the lower spectrograms come from D1 and D2 populations. Also, the x-axes are missing scales and labels.

- Figure 6. Panels A have been already shown in Fig. 5. A reorganization of the figures could also solve this kind of repetitions. The labels of Panels B, C and D, as well as the legend of Panels D, are too small to be readable.

- Line 201. Looking at the slow oscillations of beta and gamma powers in Panel Dii, it seems that the main driving frequency is lower than 5 Hz and closer to 3 Hz.

- Line 269. e.g., DAergic

- Line 361. Theta should be used as a symbol.

- Line 364. Bursting regime

- References should be double-checked since there are some formatting errors (e.g., [1] with upper case author names, some references with DOIs some without, different formats to state the publication month, etc.)

Reviewer #3: This is a very well-written modeling study that demonstrates the functional interaction of several distinct rhythmic dynamics in the thalamus, and the role of dopamine in regulating this complex systems-level interaction.

Single-cell and network simulations at increasing levels of complexity are presented, with the effects of different parameter choices or simulated perturbations clearly demonstrated at each stage. These scenarios are thoughtfully chosen to provide insight into the mechanisms underlying the dynamics. Finally, the experimental grounding for modeling decisions is clearly specified and supported by references when appropriate, or also supported by references to past modeling literature.

General comments:

1. The conclusions regarding the role of the demonstrated high DA state in switching between cell assemblies or motor programs would be significantly stronger if this switching behavior was directly demonstrated in some way. The full complexity of generating multiple distinct stable assembly states on top of the existing oscillatory dynamics would of course be well beyond the scope of this work and constitute an entire separate paper. However, perhaps some measures of the sensitivity of the existing network to perturbations during the beta-rhythm inhibition could provide a quantitative demonstration of the concept.

2. A bit more information on the choice of network-level parameters in the model would be helpful. The discussion of the D-current conductance in “Caveats and limitations” is quite comprehensive and useful, but a bit of the same logic for how the parameters for effects of dopamine were tuned in the model would be helpful. Did the magnitude and balance of effects on DA receptors and gap junctional conductance need to be tuned to create the phenomena of interest? Such hand-tuning should not detract from the conclusions presented, but a few sentences explaining the approach would be helpful for interpretation and to guide future modeling work.

3. A note on the possible effects of parameter heterogeneity on the phenomena shown would be helpful, especially in the context of the FSI network theta rhythm robustness. Does the robustness to noise conferred by network gap junction coupling hold up even if when there is heterogeneity in the intrinsic properties of the cells?

4. The authors mention that the model files are publicly available, but I was unable to locate them within the DynaSim github page. A more direct reference should be provided before publication.

5. In Figure 5, the rhythmicity of the SPN network looks quite similar between i and iii, as if the frequency preference mechanism for the low DA spiking is present in the subthreshold dynamics as well (although the frequency of the peak is slightly shifted between the subthreshold case in i and the spiking case in iii). This also means that the beta peak is significantly shifted between iii and iv. If the beta dynamics are significantly different between the low and high DA cases (in addition to the theta coupling shown), does this have other consequences for our interpretation of this state?

Minor points:

L66: commas around “stuttering and ga)mma resonance”

L126: in some spots the term “LFP surrogates” is used to convey the fact that this mean voltage is a strong simplification of LFP. That should be mentioned here, where it is first introduced

L171-176: A cross-reference to the “Caveats and limitations” section would be useful to the reader when introducing some of the stronger assumptions of the network model

**Have all data underlying the figures and results presented in the manuscript been provided?**

Reviewer #1: Yes

Reviewer #2: No: Data availability does not comply with PLOS Computational Biology Policies (see "Comments to the Authors")

Reviewer #3: Yes

PLOS authors have the option to publish the peer review history of their article (what does this mean?). If published, this will include your full peer review and any attached files.

Reviewer #1: No

Reviewer #2: Yes: Alberto Antonietti

Reviewer #3: Yes: Thomas Chartrand

---

## [Editor Report · Decision Letter 1]

19 Dec 2019

Dear Dr Chartove,

We are pleased to inform you that your manuscript 'A biophysical model of striatal microcircuits suggests delta/theta-rhythmically interleaved gamma and beta oscillations mediate periodicity in motor control' has been provisionally accepted for publication in PLOS Computational Biology.

In the meantime, please log into Editorial Manager at https://www.editorialmanager.com/pcompbiol/, click the "Update My Information" link at the top of the page, and update your user information to ensure an efficient production and billing process.

One of the goals of PLOS is to make science accessible to educators and the public. PLOS staff issue occasional press releases and make early versions of PLOS Computational Biology articles available to science writers and journalists. PLOS staff also collaborate with Communication and Public Information Offices and would be happy to work with the relevant people at your institution or funding agency. If your institution or funding agency is interested in promoting your findings, please ask them to coordinate their releases with PLOS (contact ploscompbiol@plos.org).

Thank you again for supporting Open Access publishing. We look forward to publishing your paper in PLOS Computational Biology.

Sincerely,

Michele Migliore

Guest Editor

PLOS Computational Biology

Lyle Graham

Deputy Editor

PLOS Computational Biology

---

## [Editor Report · Acceptance letter]

18 Feb 2020

PCOMPBIOL-D-19-01258R1 

A biophysical model of striatal microcircuits suggests delta/theta-rhythmically interleaved gamma and beta oscillations mediate periodicity in motor control

Dear Dr Chartove,

I am pleased to inform you that your manuscript has been formally accepted for publication in PLOS Computational Biology. Your manuscript is now with our production department and you will be notified of the publication date in due course.

With kind regards,

Laura Mallard
